# Kidins220 regulates the development of B cells bearing the λ light chain

Anna-Maria Schaffer[1,2,3], Gina Jasmin Fiala[1,2,3], Miriam Hils[1,3,4], Eriberto Natali[5], Lmar Babrak[5], Laurenz Alexander Herr[1,2,3], Mari Carmen Romero-Mulero[1,6], Nina Cabezas-Wallscheid[6,7], Marta Rizzi[3,7,8,9], Enkelejda Miho[5,10,11], Wolfgang WA Schamel[1,2,3], Susana Minguet[1,2,3]*

[1]Faculty of Biology, Albert-Ludwigs-University of Freiburg, Freiburg, Germany; [2]Signalling Research Centers BIOSS and CIBSS, University of Freiburg, Freiburg, Germany; [3]Center of Chronic Immunodeficiency CCI, University Clinics and Medical Faculty, Freiburg, Germany; [4]Department of Dermatology and Allergy Biederstein, School of Medicine, Technical University of Munich, Munich, Germany; [5]Institute of Medical Engineering and Medical Informatics, School of Life Sciences, FHNW 15 University of Applied Sciences and Arts Northwestern Switzerland, Muttenz, Switzerland; [6]Max Planck Institute of Immunobiology and Epigenetics, Freiburg, Germany; [7]CIBSS – Centre for Integrative Biological Signalling Studies, University of Freiburg, Freiburg, Germany; [8]Division of Clinical and Experimental Immunology, Institute of Immunology, Center for Pathophysiology, Infectiology and Immunology, Medical University of Vienna, Vienna, Austria; [9]Department of Rheumatology and Clinical Immunology, University Medical Center Freiburg, Faculty of Medicine, University of Freiburg, Freiburg, Germany; [10]aiNET GmbH, Basel, Switzerland; [11]SIB Swiss Institute of Bioinformatics, Lausanne, Switzerland

*For correspondence:
susana.minguet@biologie.uni-freiburg.de

**Abstract** The ratio between κ and λ light chain (LC)-expressing B cells varies considerably between species. We recently identified Kinase D-interacting substrate of 220 kDa (Kidins220) as an interaction partner of the BCR. *In vivo* ablation of Kidins220 in B cells resulted in a marked reduction of λ LC-expressing B cells. Kidins220 knockout B cells fail to open and recombine the genes of the *Igl* locus, even in genetic scenarios where the *Igk* genes cannot be rearranged or where the κLC confers autoreactivity. *Igk* gene recombination and expression in Kidins220-deficient B cells is normal. Kidins220 regulates the development of λ LC B cells by enhancing the survival of developing B cells and thereby extending the time-window in which the *Igl* locus opens and the genes are rearranged and transcribed. Further, our data suggest that Kidins220 guarantees optimal pre-BCR and BCR signaling to induce *Igl* locus opening and gene recombination during B cell development and receptor editing.

## Editor's evaluation

This manuscript addresses an important question and provides some clear instructions on the use of the λ chain locus. In particular, it highlights the role of Kidins220 in extending B cell survival so as to increase the time available for the λ light chain locus to open for VJ recombination, it also shows that extended survival time alone is not sufficient to enhance λ light chain recombination and this represents an important contribution to our understanding of B cell repertoire. The data are convincing although some mechanistic questions remain.

## Introduction

Antigen recognition in B cells is mediated by the B cell antigen receptor (BCR), composed of two immunoglobulin (Ig) heavy chains (HC) and two Igκ or Igλ light chains (LC). They form the BCR complex together with the associated Igα and Igβ heterodimer to transmit signals for B cell development, proliferation, survival, and activation. Each B cell expresses a BCR with a given specificity, which is acquired by a progressive rearrangement of the *variable* (V), *joining* (J) and, in case of the HC, *diversity* (D) gene segments of the BCR *HC* and *LC* loci in the bone marrow (BM) (reviewed in *Clark et al., 2014*; *Herzog et al., 2009*). The key enzymes that facilitate V(D)J recombination are recombination-activating gene 1 (RAG1) and RAG2 proteins (*Oettinger et al., 1990*; *Schatz et al., 1989*). V(D)J recombination is initiated at the pro-B cell stage (*Rolink et al., 1999*). An in-frame rearrangement at the *HC* genes leads to the expression of a µHC that is binding to the λ5 and VpreB components of the surrogate light chain (*Karasuyama et al., 1990*; *Reth et al., 1985*; *Tsubata and Reth, 1990*). Together with the Igα/β signaling subunits, these chains form the pre-BCR complex, which is first expressed on large pre-B cells. Signals from the pre-BCR and the interleukin (IL)–7 receptor (IL-7R) induce a proliferative burst that is followed by cell cycle attenuation, promoting *LC V-* to *J*-gene segment rearrangement in the small pre-B cell stage (*Flemming et al., 2003*; *Ma et al., 2010*; *Mandal et al., 2009*; *Ochiai et al., 2012*; *Reth et al., 1985*; *Reth and Nielsen, 2014*). A productive rearrangement leads to the pairing of the pre-existing HC with the newly generated LC, forming the IgM-BCR expressed first on the immature B cells in the BM (reviewed in *Herzog et al., 2009*; *Matthias and Rolink, 2005*).

*LC* locus opening and recombination during B cell development and receptor editing (a process changing the specificity of the BCR by secondary *LC VJ*-gene rearrangements) is dependent on transcription factors including Ikaros, Aiolos, interferon regulatory factor (IRF)–4, IRF-8 and E2A (*Bai et al., 2007*; *Beck et al., 2009*; *Inlay et al., 2004*; *Johnson et al., 2008*; *Lazorchak et al., 2006*; *Lu et al., 2003*; *Ma et al., 2006*; *Ma et al., 2008*; *Mandal et al., 2009*; *Pathak et al., 2008*; *Quong et al., 2004*; *Stadhouders et al., 2014*). Genetic mouse models individually lacking some of these transcription factors show impaired *LC* locus opening, affecting λLC expression more severely (*Beck et al., 2009*; *Lu et al., 2003*; *Pathak et al., 2008*; *Quong et al., 2004*). The activity of these transcription factors is directly or indirectly regulated by pre-BCR and BCR signaling. Mice lacking a signaling competent pre-BCR or BCR, or lacking signaling molecules downstream of the pre-BCR or BCR including the adapter protein SLP65, Bruton's tyrosine kinase (BTK) and phospholipase Cγ2 (PLCγ2), are characterized by an altered LC expression (*Bai et al., 2007*; *Dingjan et al., 2001*; *Hayashi et al., 2004*; *Kersseboom et al., 2006*; *Xu et al., 2007*). Specifically, mice deficient in SLP65 show reduced *LC* germline transcripts especially from the *Igl* locus (*Hayashi et al., 2004*; *Kersseboom et al., 2006*; *Stadhouders et al., 2014*). Likewise, the absence of BTK, a kinase that is recruited by SLP65, severely reduces *Igl* germline transcripts and LC expression (*Dingjan et al., 2001*; *Kersseboom et al., 2006*; *Stadhouders et al., 2014*). The concurrent ablation of both SLP65 and BTK abrogates *Igk* and *Igl* germline transcription and BCR surface expression (*Kersseboom et al., 2006*; *Stadhouders et al., 2014*). PLCγ2 is recruited to SLP65, regulating the calcium signaling downstream of the pre-BCR and BCR (*Hashimoto et al., 2000*; *Xu et al., 2007*). PLCγ2-knockout (KO) mice show a strong reduction of λLC B cells and a mild reduction of *Igk* germline transcripts (*Bai et al., 2007*; *Xu et al., 2007*). The combined deficiency of BTK and PLCγ2 almost completely abrogated LC expression (*Xu et al., 2007*).

Pre-BCR signaling is important for the induction of *LC VJ*-gene rearrangement at both *LC* loci (*Igk* and *Igl*). However, the ratio between the usage of the two LCs diverges greatly among species (*Sun et al., 2013*) and the regulation of their differential expression is still not fully understood. The primary B cell repertoire in mice is dominated by the κLC that is roughly ten to 20 times more frequent than the λLC (*McGuire and Vitetta, 1981*; *Sun et al., 2013*). It is generally accepted that λLC B cell generation is favored when rearrangement at the *Igk* locus is unsuccessful, or when the immature κLC-containing BCR confers autoreactivity (*Luning Prak et al., 2011*; *Nemazee, 2006*). In the latter case, the BCR specificity is modified by receptor editing. The ratio of κ/λ LC is further impacted by the survival of developing B cells. Extending the life-span of B cell precursors by overexpression of the anti-apoptotic protein B cell lymphoma 2 (BCL2), promotes the generation of λLC B cells (*Ait-Azzouzene et al., 2005*; *Derudder et al., 2009*; *Dingjan et al., 2001*). In line with this, limiting B cell survival by genetically abrogating the NF-κB signaling pathway, results in mice with a reduced amount of λLC B cells (*Derudder et al., 2009*). Based on these observations, the generation of λLC B cells

mainly depends on (i) the ability and kinetics of the *LC* locus opening, (ii) the life-span of developing B cells, and (iii) the efficiency of receptor editing during tolerance induction (*Derudder et al., 2009*).

In this study, we investigated the differential regulation of the κ- *versus* λ LC expression. We have identified the transmembrane protein Kidins220 as a new binding partner of the BCR (*Fiala et al., 2015*). Kidins220 was first described as Kinase D-interacting substrate of 220 kDa in neuronal cells (*Iglesias et al., 2000*; *Kong et al., 2001*). As a scaffold protein, Kidins220 is implicated in multiple cellular processes like survival, proliferation and receptor signaling, among which are also the antigen receptors of T and B cells (*Deswal et al., 2013*; *Fiala et al., 2015*; *Neubrand et al., 2012*). The complete genetic deletion of Kidins220 is embryonically lethal (*Cesca et al., 2012*). Conditional mb1Cre-mediated B cell specific Kidins220 KO mice (B-KO) showed reduced BCR signaling, and almost complete loss of λ LC B cells in the BM and periphery, with only mild effects on the κLC compartment (*Fiala et al., 2015*). Our new findings presented here indicate that Kidins220 is crucial for the generation of λ LC B cells by supporting B cell progenitor survival and pre-BCR and BCR signaling, which allows for *Igl* locus opening, recombination, and protein expression.

## Results

### Kidins220 B-KO mice show a skewed primary BCR repertoire

Despite almost normal B cell numbers, B-cell-specific Kidins220 KO (B-KO) mice show an approximately 75% reduction of B cells carrying a λ LC-containing IgM-BCR on the cell surface (*Figure 1A*; *Fiala et al., 2015*). To further understand the molecular mechanism leading to this phenotype, we performed in-depth analysis of the primary IgM-BCR repertoire of control (CTRL) and B-KO mice. We FACS-sorted immature (B220⁺IgM⁺IgD⁻) B cells from the BM of a pool of three individual mice per genotype and performed sequencing analyzing paired (HC and LC), full-length V(D)J sequences from cDNA of single cells. Successful purification of immature B cells was confirmed by the annotation of 98–99% of HCs to the IgM isotype (*Figure 1B*). In the CTRL, 83% of all μHCs were co-expressed with the κLC, and the remaining 17% with various subclasses of the λ LC (*Figure 1C and D*). In contrast, in B-KO B cells, the μHC was almost exclusively co-expressed with the κLC (98%) and only 2% of all cells contained a λ LC. Thus, the absence of λ LC BCRs on the surface of B cells in B-KO mice reflects absent production of mRNA of *Igl* from all λ LC subclasses.

We next investigated whether Kidins220 plays a role in the usage of specific *V-* and *J*-genes. The murine *Igh* locus consists of more than 100 *Igh-V-*, 8–12 *Igh-D-* and 4 *Igh-J*-genes (*Kenter et al., 2021*). We compared the use of each gene segment between CTRL and B-KO cells by performing a Pearson correlation test. We obtained very strong correlations for *Igh-V-* and *J*-gene usage ($r=0.97$ and $0.98$, respectively), suggesting that Kidins220 does not influence which V(D)J-gene is used for the μHC of the BCR (*Figure 1—figure supplement 1A*). The murine *Igk* locus consists of at least 101 functional *Igk-V*-genes as well as four functional *Igk-J*-genes (*Aoki-Ota et al., 2012*; *Figure 1—figure supplement 1B*). Likewise, we observed a good correlation of the *Igk-V-* and *J*-gene usage ($r=0.96$ and $r=1$, respectively) between CTRL and Kidins220-deficient cells (*Figure 1—figure supplement 1A*), confirming a rather intact κLC repertoire. Still, B-KO B cells showed a slightly increased use of *Igkj5* (14.2%) when compared to CTRL (10%) (*Figure 1E*). Use of *Igkj5* correlates with increased secondary *Igk-V-* to *J*-gene rearrangements. It frequently occurs before deletion of part of the *Igk* locus via a recombining sequence recombination and opening of the *Igl* locus (*Prak et al., 1994*; *Retter and Nemazee, 1998*). The murine *Igl* locus comprises one cassette including *Iglv2* and *Iglv3* upstream of *Iglj2-c2* (λ 2), and a second cassette containing *Iglv1* upstream of *Iglj3-c3* (λ 3) and *Iglj1-c1* (λ 1) (*Figure 1—figure supplement 1B*; *Gerdes and Wabl, 2002*; *Sanchez et al., 1996*). In mature wild type B cells, the different λ LCs are used at a frequency of 62%, 31% and 7% for λ 1, λ 2 and λ 3, respectively (*Boudinot et al., 1994*; *Sanchez et al., 1996*). This pattern was confirmed by our sequencing results for immature B cells in CTRL mice (*Figure 1D*). The *Igl-V*-gene usage showed a preference for *Iglv1* over *Iglv3* or *Iglv2*, independent of the presence of Kidins220 (*Figure 1E*). However, Kidins220-deficient B cells showed a relative increase of *Iglv1*-gene usage (88% *vs* 59% in CTRL) (*Figure 1E*). Both CTRL and B-KO immature B cells preferentially used the *Iglj1*-gene segment (44% and 69%, respectively). However, B-KO B cells used *Iglj3* more frequently than *Iglj2*. These observations suggest that in the absence of Kidins220, the genes of the second λ LC cassette are favored (*Figure 1E*). Indeed, the Pearson correlation coefficient for the *Igl-J*-gene segments was

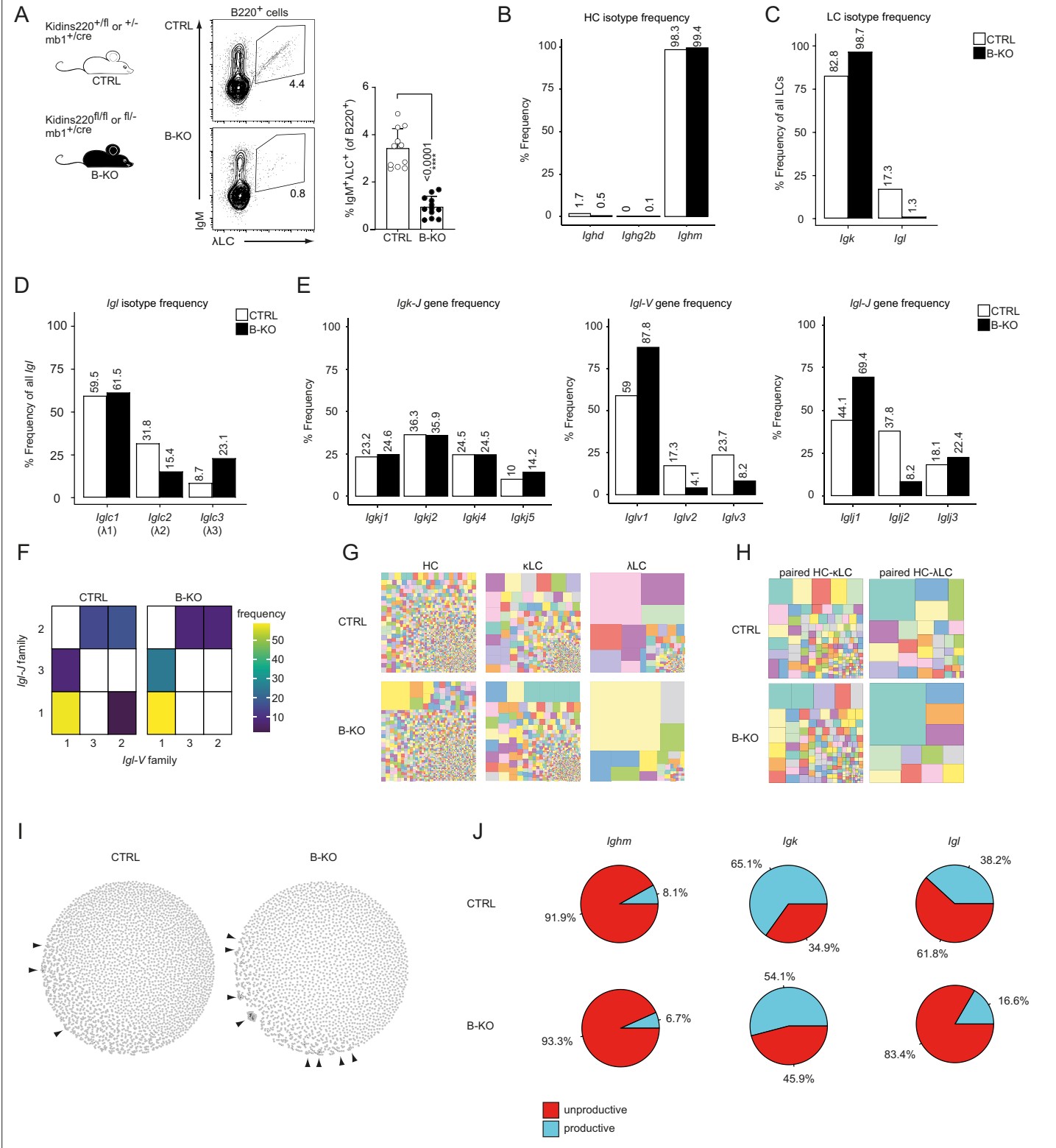

**Figure 1.** Kidins220 immature B-KO cells show a skewed primary BCR repertoire. (**A**) Schematics of the genotypes of the *Kidins220* locus and cre-recombinase expression for control (CTRL) and B cell specific knockout (B–KO) mice (left). Representative flow cytometric BM analysis using antibodies for B220, IgM and λ LC as well as statistical quantification (right, five independent experiments were pooled; n=11–12 mice per genotype). The mean + SD is plotted. Statistical analysis was performed using unpaired Student's *t*-test. p-value is indicated. (**B–J**) Immature B cells (B220⁺IgM⁺IgD⁻) of three individual mice per genotype were pooled and subjected to single cell sequencing analyzing full-length *Ig*-gene V(D)J recombination status and BCR

*Figure 1 continued on next page*

*Figure 1 continued*

repertoire based on cDNA. More than 10,000 cells per genotype were analyzed. Only productive recombinations are used for the analyses in (**B–I**). The frequencies of the individual HCs (**B**) and LCs (**C**) isotypes in CTRL and B-KO of all obtained sequences are plotted. (**D**) Relative frequencies of the individual λ LC families are depicted. (**E**) The Frequencies of *V*- and *J*-gene sequences of the *Igk* and *Igl* loci of all obtained *V*- or *J*-gene sequences within the respective LC isotype are plotted. (**F**) Heatmap of the *Igl-V-J* gene combinations of the λ LC. (**G**) Tree maps illustrating clonal CDR3 frequency of HC, κ LC, and λ LC. Each square represents an individual CDR3, and its frequency correlates with the square size. (**H**) Tree maps illustrating the clonal CDR3 frequency of the whole BCR (paired HC-LC). Each square represents an individual CDR3, and its frequency correlates with the square size. (**I**) Primary antibody repertoire networks showing unique CDR3 sequences (nodes) as individual dots. Dots are connected via similarity edges only when their sequence differs in just one amino acid. Clustered dots indicative of clonal expansion are indicated with arrowheads. (**J**) Productive and unproductive read frequencies.

The online version of this article includes the following source data and figure supplement(s) for figure 1:

**Source data 1.** Kidins220 immature B-KO cells show a skewed primary BCR repertoire (*Figure 1* and *Figure 1—figure supplement 1*).

**Figure supplement 1.** Immature B-KO B cells show a skewed primary BCR repertoire.

weak in B-KO cells (*r*=0.51) (*Figure 1—figure supplement 1A*). In the *Igl* locus, *VJ*-gene recombination preferentially takes place within the same cassette and is almost absent between cassettes (*Reilly et al., 1984*; *Sanchez et al., 1991*). In line with this, we almost exclusively detected recombination of *Iglv1* to either *Iglj3* or *Iglj1*, whereas *Iglv2* and *Iglv3* recombined to *Iglj2* (*Figure 1F*). Both CTRL and Kidins220-deficient λ LC B cells showed a prominence of *Iglv1- Iglj1* joins. We next compared the HC, κLC and λ LC repertoires of CTRL and B-KO immature B cells by analyzing the frequency of unique CDR3s in all chains. Both genotypes showed a clear polyclonal distribution for the κLC, but polyclonality was slightly reduced for the HC and strongly restricted for the λ LC in the B-KO immature B cells (*Figure 1G*). The analysis of paired HC-LC combinations highlighted that indeed the repertoire of those BCR bearing a λ LC was strongly restricted in the B-KO immature B cells (*Figure 1H*). We did not observe any predispositions for potentially autoreactive BCRs as characterized by longer or more positively charged CDR3s within their variable domain (*Figure 1—figure supplement 1C, D*).

We further analyzed the architecture of the BCR repertoire by applying similarity networks (*Figure 1I*; *Miho et al., 2019*). Briefly, we created networks in which unique CDR3 sequences (nodes) are represented as individual dots. These nodes are connected via similarity edges only if their sequence differs by one amino acid (Levenshtein distance = 1). Thus, similar CDR3 sequences appear clustered indicating clonal expansion and are highlighted by arrowheads (*Figure 1I*). Unexpectedly, the primary B cell repertoire of B-KO mice showed signs of specific clonal expansion, especially of one single clone containing the *Ighv10-1*-gene segment (*Figure 1I*). The usage of this *Igh-V*-gene segment has been previously associated with anti-DNA antibodies and herpesvirus infections (*Maranhão et al., 2013*; *Zelazowska et al., 2020*). Lastly, we determined the ratio between productive and unproductive rearrangements (*Figure 1J*). Unproductive sequences are defined as out-of-frame sequences, sequences containing premature stop codons, orphon genes or non-*Ig* sequences (*Smakaj et al., 2020*). They encompass a significant high proportion of the raw outputs (*Smakaj et al., 2020*) and are usually removed during a preprocessing step prior to data analysis (as done for *Figure 1B–I* and *Figure 1—figure supplement 1*). Both, CTRL and B-KO, showed similar frequencies of sequences defined as unproductive rearrangements (92% and 93%, respectively) from the *Igh* locus. In contrast, the frequency of sequences defined as unproductive rearrangements from the *Igl* and *Igk* loci was increased in B-KO compared to CTRL cells: 1.2 times for *Igk*, and 2 times for the *Igl* locus (*Figure 1J*). Deeper investigations revealed that these unproductive rearrangements within the *Igl* locus were caused by a series of premature stop-codons. Taken together, genetic ablation of Kidins220 in B cells skews the primary BCR repertoire mainly due to unsuccessful production of λ LCs.

## Kidins220 is essential for the opening of the *Igl* locus

Kidins220-deficient B cells failed to express λ LC. The expression of the κLC and λ LC depends on at least three factors: (i) the ability and kinetics of the *Igk* and *Igl* loci opening, (ii) the life-span of the pre-B cells, and (iii) the level of receptor editing during tolerance induction (*Dingjan et al., 2001*). Hence, we generated BM-derived pro-/pre-B cell cultures from CTRL and B-KO mice BM to study their ability to open the *LC* loci (*Figure 2A*). After 7 days in the presence of IL-7, we obtained a population almost homogeneous for surface expression of B220 and lacking a surface BCR, indicative of pro- and pre-B cells (data not shown *Rolink et al., 1991*). Subsequent IL-7 withdrawal led to a relative increase

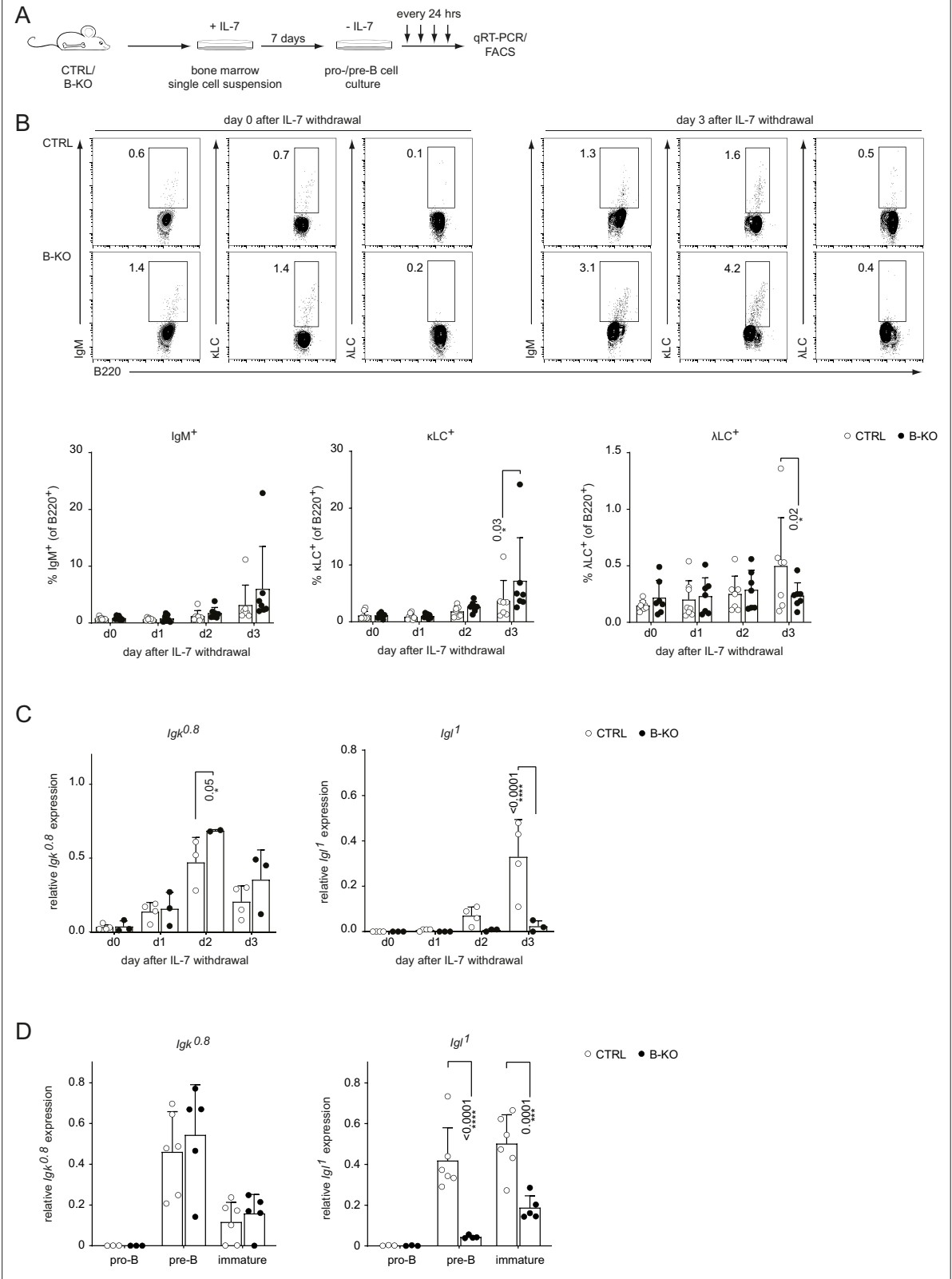

**Figure 2.** Kidins220 is required for the opening of the *Igl* locus in pro-/pre-B cells. (**A**) Experimental setup to generate primary BM derived pro-/pre-B cell cultures. Total BM was isolated from CTRL and B-KO mice and cultured for 7 days in medium supplemented with IL-7. Next, IL-7 was removed, and samples were taken immediately (day 0) and every 24 hr for subsequent analyses. (**B**) Flow cytometry assayed surface expression of B220, IgM, λ LC, and κ LC. Representative plots of day 0 and day 3 as well as statistical analyses are shown (n=7–8 per genotype). (**C, D**) Isolated RNA was reverse transcribed

*Figure 2 continued on next page*

*Figure 2 continued*

to assay the relative expression of germline transcripts of the *Igk* and *Igl* loci by qRT-PCR. (**C**) Normalization was done using *Hrpt* (n=2–4 per genotype). (**D**) Pro-B cells (B220$^+$CD117$^+$CD25$^-$IgM$^-$), pre-B cells (B220$^+$CD117$^-$CD25$^+$IgM$^-$) and immature B cells (B220$^+$IgM$^+$IgD$^-$) were FACS sorted from CTRL and B-KO mice. Normalization was done using *Hrpt* and *Actb* (n=3–6 per genotype). In all graphs, the mean + SD is plotted. Each symbol represents one mouse. Statistical analysis was performed using Two-way ANOVA with Fisher's (Least Significant Difference) LSD test. Only p-values <0.05 are indicated.

The online version of this article includes the following source data and figure supplement(s) for figure 2:

**Source data 1.** Kidins220 is required for the opening of the Igl locus in pro-/pre-B cells.

**Figure supplement 1.** The transcriptional network controlling LC recombination as well as RAG protein activity are independent of Kidins220.

**Figure supplement 1—source data 1.** Kidins220 is required for the opening of the Igl locus in pro-/pre-B cells.

of cells expressing a BCR on their surface (*Dingjan et al., 2001*; *Middendorp et al., 2002*). CTRL and B-KO showed increasing levels of κLC$^+$IgM-BCR$^+$ B cells over time (*Figure 2B*). The proportion of $\lambda$ LC$^+$IgM-BCR$^+$ B cells was higher in CTRL compared to B-KO B cells at day 3 of culture, suggesting that the *in vivo* phenotype is B cell intrinsic.

Prior to *LC VJ*-gene rearrangement, the gene locus is opened and germline transcripts are produced (also known as sterile transcripts that do not encode for any protein but are proposed to possess regulatory functions) (*Engel et al., 1999*; *Schlissel and Baltimore, 1989*). We analyzed the relative expression of germline transcripts from the *Igk* and *Igl* loci in CTRL and B-KO pro-/pre-B cell cultures and in directly sorted pro-, pre- and immature B cells by qRT-PCR (*Figure 2C and D*; *Figure 2—figure supplement 1A*). The relative amount of *Igk* germline transcript and the kinetics of induction was similar between CTRL and B-KO B cells. However, Kidins220-deficient B cells failed to induce germline transcription from the *Igl* locus to the levels of CTRL cells in pro-/pre-B cell cultures upon IL-7 withdrawal and in sorted pre- and immature B cells (*Figure 2C and D*). The *Igk* germline transcripts in the CTRL cells peaked at day 2 compared to day 3 for the *Igl* germline transcripts, in line with the findings that activation of the *Igk* locus precedes the *Igl* locus (*Engel et al., 2001*; *Luning Prak et al., 2011*; *Nemazee, 2006*). *LC* germline transcription is dependent on the expression of E2A proteins (E12 and E47), which in turn regulate the RAG1 and RAG2 proteins that are responsible for VJ-gene recombination (*Beck et al., 2009*; *Hsu et al., 2003*; *Quong et al., 2004*). We did not detect major differences in the expression of *Tcf3* (encodes for *E12* and *E47*), *Rag1* and *Rag2* mRNA as assayed by qRT-PCR between CTRL and B-KO B cells shortly upon IL-7 withdrawal (*Figure 2—figure supplement 1B*) nor in directly sorted cells (*Figure 2—figure supplement 1C*). Next, we explored the possibility that Kidins220 regulates RAG1 and Rag2 activity. To this end, we used a RAG-activity reporter (GFPi) harboring an inverted GFP sequence flanked by RAG-recognition signal sequences and an in-frame RFP sequence for tracking positively transduced cells (*Trancoso et al., 2013*). Thus, RAG activity can be monitored by the generation of GFP$^+$ cells. We transduced CTRL and B-KO pro-/pre-B cell cultures and withdrew IL-7 2 days later. The proportion of GFP$^+$ cells over the 3 days upon IL-7 withdrawal was comparable between CTRL and B-KO cells. This suggests that RAG protein activity *per se* is not affected by the absence of Kidins220 (*Figure 2—figure supplement 1D*). Together, these data suggest that Kidins220 is specifically necessary for *Igl* locus opening and/or its transcription in developing B cells. Mechanistically, these effects are independent of the transcriptional or functional regulation of elements of the recombination machinery.

## Kidins220 is required for the generation of $\lambda$ LC B cells in κ-KO mice

We next aimed to force the development of $\lambda$ LC B cells in a Kidins220-deficient background by crossing CTRL and B-KO mice to κ-deficient mice. We used the well-described iEκT model, where the intronic κ enhancer (iEκ) is replaced by a neomycin resistance cassette leading to a silenced *Igk* locus (from now on indicated as κ-KO). Consequently, all developing B cells exclusively express the $\lambda$ LC (*Takeda et al., 1993*). We used CTRL and B-KO mice heterozygous for the iEκT allele as littermate controls (κ-CTRL). We first analyzed the B cell compartment in the BM of all four genotypes (*Figure 3—figure supplement 1A*). As shown in *Figure 3A–H* and *Figure 3—figure supplement 1B–F*, the heterozygous ablation of one of the $\kappa$ alleles did not overall change the already described phenotype of CTRL and B-KO mice (*Fiala et al., 2015*). B-KO mice showed B cell frequencies and cell numbers comparable to CTRL but a severe reduction of $\lambda$ LC$^+$ B cells (*Figure 3—figure supplement 1B, F*; *Figure 3A–C,H*). In both CTRL and B-KO mice, κ-KO did not significantly alter the total amount

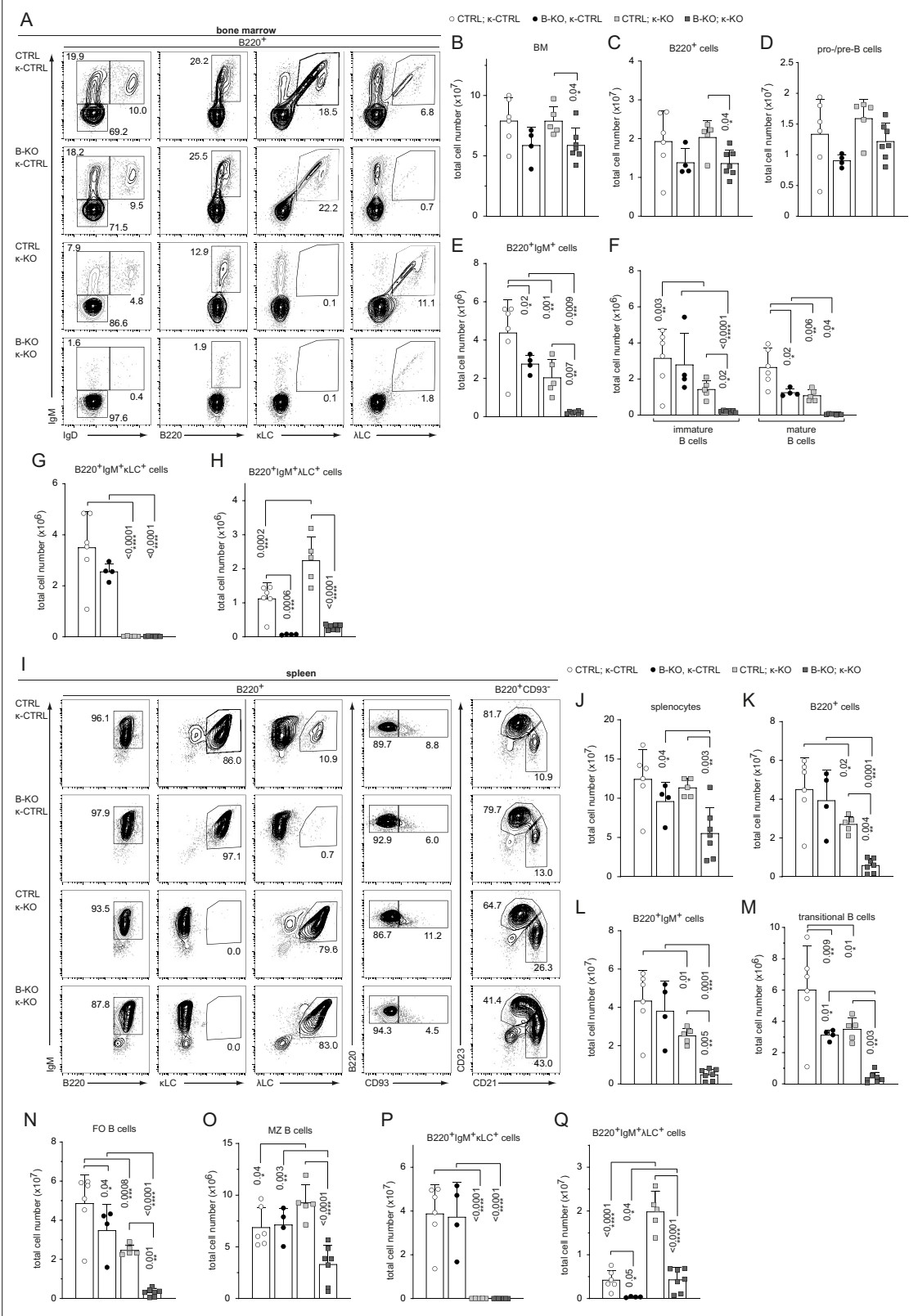

**Figure 3.** Kidins220 is required for the generation of λ LC-bearing B cells even in κ-KO mice. (**A, I**) Representative flow cytometric analysis of the BM (**A**) and spleen (**I**) of CTLR and B-KO mice in κ-sufficient (κ-CTRL) and -deficient (κ-KO) backgrounds. (**B-H; J–Q**) Total cell numbers for each B cell compartment from BM (**B–H**) and spleen (**J–Q**). Pro-/pre-B cells are defined as B220$^+$IgM$^-$IgD$^-$, immature B cells as B220$^+$IgM$^+$IgD$^-$, mature B cells as B220$^+$IgM$^+$IgD$^+$, transitional B cells as B220$^+$CD93$^+$, follicular (FO) B cells as B220$^+$CD93$^-$IgM$^+$CD21$^{med}$CD23$^{high}$ and marginal zone (MZ) B cells as

*Figure 3 continued on next page*

*Figure 3 continued*

B220$^+$CD93$^-$IgM$^+$CD21$^{high}$CD23$^{med}$. For the quantification, three independent experiments with n=4–7 mice per genotype were pooled. Each symbol represents one mouse. In all graphs, the mean + SD is plotted. Statistical analysis was performed using Two-way ANOVA (**F**) or One-way ANOVA with Fisher's LSD test. Only p-values <0.05 are indicated.

The online version of this article includes the following source data and figure supplement(s) for figure 3:

**Source data 1.** Kidins220 is required for the generation of $\lambda$ LC-bearing B cells even in $\kappa$ -KO mice.

**Figure supplement 1.** Kidins220 is required for the generation of $\lambda$ LC bearing B cells even in $\kappa$ -KO mice.

**Figure supplement 1—source data 1.** Kidins220 is required for the generation of $\lambda$ LC bearing B cells even in $\kappa$ -KO mice.

of B220$^+$ B cells in the BM (*Figure 3B and C*). However, a severe developmental block was observed in the generation of BCR$^+$ B cells in both κ-KO genotypes shown by a decreased proportion of IgM$^+$ (immature and mature) B cells and a corresponding increase in pro-/pre-B cells (*Figure 3—figure supplement 1C*). The total pro-/pre-B cell numbers remained unchanged (*Figure 3D*). Remarkably, the reduction in IgM$^+$ cells in κ-KO mice was stronger in B-KO mice, representing only 15% of the B220$^+$IgM$^+$ population of the CTRL κ-KO mice (*Figure 3—figure supplement 1D*; *Figure 3E, F*). As expected, neither CTRL nor B-KO mice generated κLC-bearing B cells in the κ-KO background (*Figure 3G*; *Figure 3—figure supplement 1E*). Instead, all generated BCR$^+$ B cells used the $\lambda$ LC (*Figure 3—figure supplement 1D, F*; *Figure 3E,H*). Indeed, this data explain the remarkable loss of IgM$^+$ cells in B-KO κ-KO double knockout mice (*Figure 3E*). As previously described, κ-KO amplified $\lambda$ LC usage in CTRL mice (*Takeda et al., 1993*). Although some amplification also occurred in Kidins220-deficient B cells, the amount of $\lambda$ LC$^+$ B cells was severely reduced compared to CTRL (*Figure 3H*; *Figure 3—figure supplement 1F*).

Immature IgM$^+$ B cells emerge from the BM and become transitional B cells, which further mature in the spleen. Although the total number of splenocytes did not significantly change between κ-sufficient and -deficient backgrounds in CTRL mice, κ-KO mice showed an overall reduction of the B220$^+$ and B220$^+$IgM$^+$ B cell percentages and numbers in the spleen (*Figure 3—figure supplement 1G, H*; *Figure 3I–L*), concomitant with a relative increase in T cells (data not shown; *Takeda et al., 1993*). This reduction in B220$^+$ B cells was strongly amplified by Kidins220-deficiency (60–80% reduction of B220$^+$ B cells in B-KO κ-KO compared to CTRL κ-KO mice; *Figure 3K*; *Figure 3—figure supplement 1G*). Silencing of the *Igκ* locus, or deletion of Kidins220 expression in B cells, diminished transitional B cell numbers and this effect was enhanced in the double knockout (*Figure 3M*; *Figure 3—figure supplement 1I*). Most of the transitional B cells further mature into follicular (FO) B cells, whereas a smaller fraction develops into marginal zone (MZ) B cells (*Pillai and Cariappa, 2009*). We previously reported that in the absence of Kidins220, maturation towards the MZ compartment is favored resulting in around 18% of all mature splenic B cells with a MZ B cell phenotype (*Figure 3J and K*; *Figure 3N, O*; *Fiala et al., 2015*). κ-KO mice additionally lacking Kidins220 showed an even stronger reduction in the FO compartment, skewing the differentiation toward the MZ compartment (*Figure 3N and O*; *Figure 3—figure supplement 1J, K*). As expected, all splenic B cells exclusively express $\lambda$ LC BCRs in κ-KO mice, independent of Kidins220 expression (*Figure 3—figure supplement 1L, M*; *Figure 3P,Q*). In the presence of Kidins220, the development and maturation of $\lambda$ LC$^+$ B cells compensates the B cell numbers reaching about 50% of the κLC-sufficient situation in most populations (*Figure 3K–N*). However, in the absence of Kidins220, $\lambda$ LC-expressing B cell numbers are significantly reduced failing compensation to control levels (*Figure 3Q*). Taken together, Kidins220 is necessary for the development of $\lambda$ LC B cells even in κ-KO mice.

## BCL2 overexpression in Kidins220-deficient B cells partially rescues λLC usage *in vitro* and *in vivo*

For the generation of $\lambda$ LC B cells, the precursors should survive long enough to reach the state when the genes of the *Igl* locus becomes rearranged since *Igk* gene rearrangement precedes *Igl VJ* recombination (*Luning Prak et al., 2011*; *Nemazee, 2006*). To explore whether Kidins220-deficiency compromises the generation of $\lambda$ LC B cells by affecting B cell survival, we determined the amount of early apoptotic and dead cells throughout B cell development in the BM of CTRL and B-KO mice. The amount of early apoptotic (Annexin V$^+$PI$^-$) and dead (Annexin V$^+$PI$^+$) cells were tendentially increased in the absence of Kidins220 during B cell development that became statistically significant for $\lambda$ LC$^+$ B

cells (*Figure 4—figure supplement 1A*). This is in line with our previous report showing that *ex-vivo* cultured pro-/pre-B cells from B-KO mice have a decreased survival upon IL-7 withdrawal compared to CTRLs (*Fiala et al., 2015*). Since developing B cells in the BM depend on functional IL-7R signaling for survival prior to pre-BCR expression, we investigated IL-7R expression in CTRL and B-KO mice. As expected, IL-7R expression was higher in pro-B cells and decreased subsequently throughout B cell maturation (*Figure 4—figure supplement 1B*). No differences were detected between CTRL and B-KO mice. Still, absence of Kidins220 resulted in enhanced IL-7R signaling via the PI3K pathway as shown by increased phosphorylation of AKT on S473 and of its target FOXO1 on S256 in both pro- and pre-B cells (*Figure 4—figure supplement 1C*).

Next, we aimed to rescue the survival defect in B-KO B cells upon IL-7 withdrawal by retrovirally transducing primary BM-derived pro-/pre-B cell cultures from CTRL and B-KO mice with a BCL2-IRES-GFP overexpression or Mock plasmid (*Figure 4A*). Indeed, BCL2 overexpression rescued B cell survival from CTRL and B-KO mice to a comparable level over 9 days following IL-7 withdrawal (*Figure 4B*). Flow cytometric analysis of *BCL2*-transduced pro-/pre-B cell cultures revealed that IL-7 withdrawal induced κLC and $\lambda$LC surface expression in both CTRL and B-KO B cells (*Figure 4C*). However, even though *BCL2*-transduced B-KO B cells showed a pronounced increase of $\lambda$LC$^+$ B cells, it was still significantly reduced compared to CTRL cells. This reduction was compensated by a slightly enlarged κLC$^+$ B cell proportion in B-KO B cells (*Figure 4C*). *BCL2*-transduced CTRL and B-KO B cells revealed a similar induction and kinetics of *Igk* germline transcripts (*Figure 4D*). In contrast, Kidins220-deficiency dampened the abundance of *Igl* germline transcripts compared to CTRLs, even though the relative abundance of *Igl* germline transcripts seemed to be partially rescued after 9 days of IL-7 withdrawal (*Figure 4D*). In addition, the abundance of *Tcf3 (E47)* and *Rag2* mRNA transcripts was significantly reduced in B-KO cultures compared to CTRL (*Figure 4E*). These findings strongly indicate that Kidins220 plays a pivotal role in regulating factors within the recombination machinery, consequently influencing LC recombination. Notably, this effect becomes apparent when the survival defect dependent on Kidins220 is rectified through BCL2 overexpression. Together, these data confirm a role for Kidins220 in the survival of B cell precursors. However, the function of Kidins220 seems to be beyond just facilitating B cell survival, since BCL2 overexpression did not fully rescue $\lambda$LC expression in Kidins220-deficient primary B cell cultures.

Next, we crossed CTRL and B-KO mice to BCL2-transgenic mice, in which the BCL2-transgene controlled by the vav-promoter results in BCL2 expression in all nucleated cells of hematopoietic origin (*Ogilvy et al., 1999*; *Figure 5—figure supplement 1A*). BCL2 overexpression increased the percentage and total number of B cells in the BM in CTRL mice (*Figure 5A–C*; *Figure 5—figure supplement 1B*) as previously reported (*Ogilvy et al., 1999*). However, this was not the case in B-KO mice (*Figure 5A–C*; *Figure 5—figure supplement 1B*). IgM$^+$IgD$^+$ mature B cells clearly accumulated in the BM of CTRL BCL2-transgenic mice as previously shown (*Vandenberg et al., 2014*; *Yabas et al., 2011*; *Figure 5—figure supplement 1C, D*; *Figure 5D,E*). The mature IgM$^+$IgD$^+$ B cell population in the BM of B-KO BCL2-transgenic mice showed a similar tendency towards expansion, although total cell numbers remained significantly lower than in CTRL mice (*Figure 5—figure supplement 1C, D*; *Figure 5D,E* ). Both the percentages and total cell numbers of both κLC$^+$ and $\lambda$LC$^+$ B cells were expanded in BCL2-transgenic CTRL as described (*Derudder et al., 2009*; *Dingjan et al., 2001*; *Figure 5F and G*; *Figure 5—figure supplement 1E, F*). In B-KO mice, BCL2-transgenic expression failed to increase the percentage and cell numbers of κLC$^+$ and $\lambda$LC$^+$ B cells to CTRL levels (*Figure 5F and G*; *Figure 5—figure supplement 1E, F*). In line with a previous report (*Ait-Azzouzene et al., 2005*), the stronger expansion of the $\lambda$LC$^+$ compartment (sixfold for CTRL and 14-fold for B-KO) when compared to the κLC$^+$ compartment (twofold for CTRL and fourfold for B-KO) evidences the strong dependency of $\lambda$LC$^+$ B cells on B cell survival and suggests a pivotal role for Kidins220 in promoting B cell survival (*Figure 5—figure supplement 1G*). While BCL2 overexpression-driven increase in B cell precursors in the BM resulted in increased total number of splenocytes, IgM$^+$, κLC$^+$ and $\lambda$LC$^+$ B cells as reported in CTRL mice (*Ogilvy et al., 1999*; *Vandenberg et al., 2014*; *Yabas et al., 2011*), this was not the case in the absence of Kidins220 (*Figure 5H–M* and *Figure 5I–P*). The percentage of B cells in the spleen was slightly reduced in both CTRL and B-KO mice because of a concomitant expansion of T cells (*Figure 5—figure supplement 1H*; data not shown). In the four genotypes analyzed, splenic B cells were IgM positive (*Figure 5—figure supplement 1I*). The amount of transitional B cells was increased in BCL2-overexpressing CTRL mice (*Figure 5N*), but their proportion within the B220$^+$ B

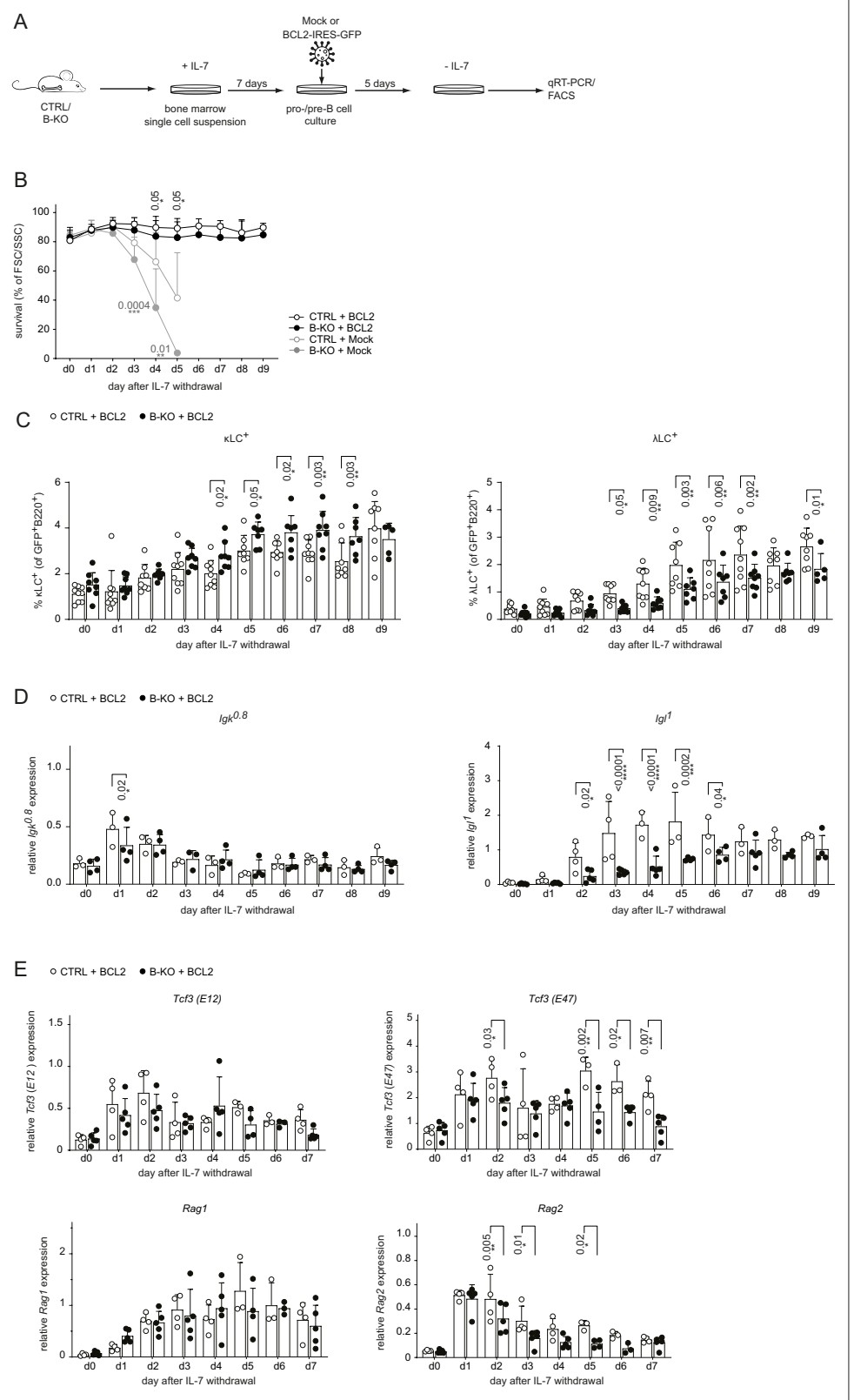

**Figure 4.** BCL2-mediated survival partially rescues $\lambda$ LC deficiency *in vitro*. (**A**) Experimental setup to generate primary BM-derived pro-/pre-B cell cultures overexpressing BCL2. Total BM was isolated from CTRL and B-KO mice and cultured for 7 days in medium supplemented with IL-7. The cells were retrovirally transduced with a BCL2-IRES-GFP overexpression or Mock plasmid, and cultured in the presence of IL-7 for 5 more days. Upon IL-7

*Figure 4 continued on next page*

*Figure 4 continued*

withdrawal, samples were collected directly (day 0) and then every 24 hr for 9 days for analysis. (**B**) The percentage of living cells was estimated using flow cytometry based on FSC/SSC daily (three to five independent experiments; n=5–11 mice per genotype). (**C**) Surface expression of $\lambda$ LC and $\kappa$ LC in *BCL2*-transduced pro-/pre-B cell cultures assayed by flow cytometry (four independent experiments; n=5–9 per genotype). (**D, E**) RNA was isolated and reverse transcribed to quantify the relative expression of (**D**) *Igk*$^{0.8}$ and *Igl*$^1$ germline transcripts (n=3–4) and (**E**) *Tcf3* (E12 and E47), *Rag1* and *Rag2* transcripts analyzed by qRT-PCR. (n=4–5 mice per genotype). Data were normalized to *Hrpt*. Each symbol represents one mouse (**C–E**). In all graphs, the mean + SD is plotted. Statistical analysis was performed using Two-way ANOVA with Fisher's LSD test. Only p-values <0.05 are indicated.

The online version of this article includes the following source data and figure supplement(s) for figure 4:

**Source data 1.** BCL2-mediated survival partially rescues $\lambda$ LC deficiency *in vitro*.

**Figure supplement 1.** Kidins220 influences B cell survival and IL-7R signaling.

**Figure supplement 1—source data 1.** Kidins220 influences B cell survival and IL-7R signaling.

cells was reduced both in CTRL and in Kidins220-deficient mice compared to non-transgenic mice (*Figure 5N*; *Figure 5—figure supplement 1J*). As previously reported, enforced BCL2 expression severely reduced the MZ B cell compartment in CTRL mice (*Banerjee et al., 2008*; *Derudder et al., 2016*) and B-KO mice (*Figure 5O and P*; *Figure 5—figure supplement 1K, L*). Most importantly, BCL2-overexpression led to a relative and absolute increase of $\lambda$ LC$^+$ B cells in both CTRL and B-KO mice (*Figure 5—figure supplement 1M–O*). For the CTRLs, this is in line with previous reports in other BCL2-transgenic mouse models (*Dingjan et al., 2001*; *Lang et al., 1997*).

Taken together, enforced survival by overexpression of the anti-apoptotic protein BCL2 enhanced the generation of $\lambda$ LC B cells in CTRL and B-KO mice. Still, BCL2 overexpression in the absence of Kidins220 failed to restore the total numbers of peripheral $\lambda$ LC B cells to control levels implying a role for Kidins220 in the generation of $\lambda$ LC B cells beyond regulating B cell survival.

## Kidins220 facilitates λLC expression during *in vivo* receptor editing

The amount of $\lambda$ LC$^+$ B cells in the repertoire is also determined by the extent of receptor editing during the establishment of tolerance (*Nemazee, 2006*). Autoreactive BCRs transmit signals upon self-antigen recognition resulting in the elimination of the autoreactive BCR from the repertoire by replacing the LC with an innocuous one by secondary *VJ*-gene rearrangement at the *Igk* or *Igl* locus. To analyze the potential role of Kidins220 during tolerance induction *in vivo*, we used a previously described model characterized by the ubiquitous transgenic expression of a membrane bound anti-κLC-reactive single chain antibody: the κ-macroself mice (*Ait-Azzouzene et al., 2005*). By using κ-macroself mice as recipients for BM reconstitutions from either CTRL or B-KO mice, all developing donor B cells expressing a κLC will recognize the ubiquitously expressed κ-transgene as a self-antigen, initiating receptor editing and generating B cells with solely $\lambda$ LC surface expression to maintain immune tolerance (*Ait-Azzouzene et al., 2005*). Hematopoietic stem cells (HSC) from CTRL and B-KO mice expressing the leukocyte marker CD45.2 were injected into sublethally irradiated CD45.1$^+$ wild type (WT) or CD45.1$^+$ κ-macroself mice (*Figure 6A*). The HSC of CTRL and B-KO mice gave rise to similar percentages of B220$^+$ B cells in the BM of both CD45.1$^+$ WT and κ-macroself transgenic mice (*Figure 6B*; *Figure 6—figure supplement 1A*), with a slight increase in total B cell numbers in κ-macroself mice (*Figure 6C and D*). κLC$^+$ B cells were rarely detected in κ-macroself mice reconstituted with either CTRL (*Ait-Azzouzene et al., 2005*) or Kidins220-deficient HSC (*Figure 6E*; *Figure 6—figure supplement 1B*). Hence, recognition of strong self-antigen signals and successful elimination of autoreactive BCRs from the cell surface does not require Kidins220. Transfer of B-KO HSC to non-transgenic WT mice still showed the reduction in $\lambda$ LC-expressing B cells in the BM compared to transfer of CTRL-HSC (*Figure 6F*; *Figure 6—figure supplement 1C*). These findings strengthen once more that the diminished production of $\lambda$ LC cells in the absence of Kidins220 is B-cell intrinsic. As expected, negative selection of κLC$^+$ B cells was accompanied by a significant increase in $\lambda$ LC usage when CTRL HSC were transferred into κ-macroself mice (*Ait-Azzouzene et al., 2005*; *Beck et al., 2009*; *Figure 6F*; *Figure 6—figure supplement 1C*). In κ-macroself mice reconstituted with HSC from B-KO mice, total $\lambda$ LC$^+$ B cell numbers were elevated compared to B-KO (*Figure 6F*). However, the percentage and total number of $\lambda$ LC$^+$ B cells were still significantly decreased compared to their control counterparts (*Figure 6F*; *Figure 6—figure supplement 1C*). The percentage and total

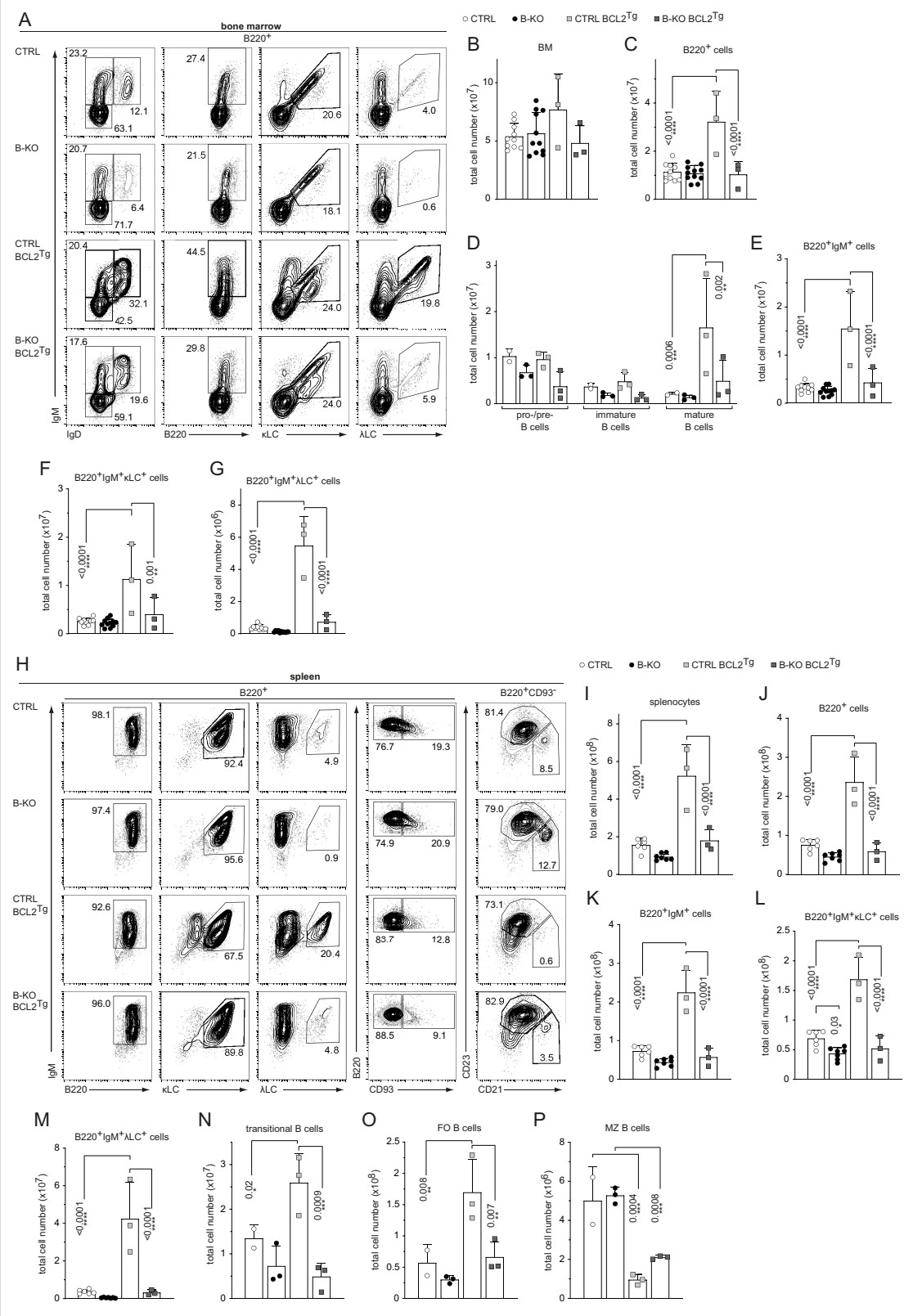

**Figure 5.** Ectopic BCL2-expression partially rescues $\lambda$ LC expression in Kidins220 B-KO mice. (**A, H**) Representative flow cytometry plots of CTRL and B-KO mice in the absence or presence of the BCL2 transgene of BM (**A**) and spleen (**H**). Total cell numbers for each B cell compartment from BM (**B–G**) and spleen (**I–P**) are shown. Pro-/pre-B cells are defined as B220$^+$IgM$^-$IgD$^-$, immature B cells as B220$^+$IgM$^+$IgD$^-$, mature B cells as B220$^+$IgM$^+$IgD$^+$, transitional B cells as B220$^+$CD93$^+$, follicular (FO) B cells as B220$^+$CD93$^-$IgM$^+$CD21$^{med}$CD23$^{high}$ and marginal zone (MZ) B cells as B220$^+$CD93$^-$

*Figure 5 continued*

IgM⁺CD21^high^CD23^med^. The quantification of three to five independent experiments with n=2–12 mice per genotype were pooled. Each symbol represents one mouse. In all graphs, the mean + SD is plotted. Statistical analysis was performed using Two-way ANOVA (**D**) or One-way ANOVA with Fisher's LSD test. Only p-values <0.05 are indicated.

The online version of this article includes the following source data and figure supplement(s) for figure 5:

**Source data 1.** Ectopic BCL2-expression partially rescues $\lambda$ LC expression in Kidins220 B-KO mice.

**Figure supplement 1.** Ectopic BCL2 expression partially rescues $\lambda$ LC expression in B-KO mice.

**Figure supplement 1—source data 1.** Ectopic BCL2 expression partially rescues $\lambda$ LC expression in B-KO mice.

numbers of BCR⁻ B cells in the BM were significantly increased in κ-macroself mice independent of Kidins220 expression (*Figure 6G*; *Figure 6—figure supplement 1D*) as previously observed (*Ait-Azzouzene et al., 2005*).

The total B cell compartment in the spleens of these mice was reduced in percentage and numbers compared to non-transgenic littermates, most probably due to the reduced output from the BM (*Figure 6H–J*; *Figure 6—figure supplement 1E*; *Ait-Azzouzene et al., 2005*; *Beck et al., 2009*). The lack of κLC-expressing B cells in the periphery in both Kindins220 sufficient and deficient cells indicated a functional tolerance induction (*Figure 6K*; *Figure 6—figure supplement 1F*). Concomitantly, there was an increase in $\lambda$ LC usage in both groups (*Figure 6—figure supplement 1G*). However, $\lambda$ LC B cell numbers were only significant increased in κ-macroself mice reconstituted with HSC from CTRL mice (*Figure 6L*). As in the BM, the relative numbers of splenic BCR⁻ B cells were increased in κ-macroself mice (*Figure 6—figure supplement 1H*), resulting in a significant increase in the total cell numbers for κ-macroself mice reconstituted with HSC from CTRL mice (*Figure 6M*). These BCR⁻ B cells might have downregulated their BCR expression below the detection limit due to its recognition of the κ-macroself antigen.

In all, these data indicate that Kidins220 is not required for the transmission of signals downstream of the IgM-BCR upon recognition of strong self-antigens such as the chimeric anti-κLC antibodies, and for the subsequent elimination of the autoreactive BCRs from the repertoire. However, Kidins220 plays a pivotal role for the efficient generation of $\lambda$ LCs that fulfill the repertoire. Consequently, in B-KO mice, there is a deficiency of B cells capable of responding to foreign epitopes recognized by BCRs carrying the $\lambda$ LC, resulting in what has been described as 'holes in the BCR repertoire' (*Goodnow, 1996*).

## Ectopic BCL2 fails to rescue $\lambda$LC expression during *in vivo* receptor editing

We next asked whether increased survival might rescue $\lambda$ LC expression during receptor editing. To this end, we modified our BM-transfer protocol by transducing the isolated HSC of CTRL and B-KO mice 24 hr post-isolation with a BCL2-IRES-GFP overexpression plasmid. Following another 24 hr of HSC culture in optimized medium, the cells were injected into sublethally irradiated CD45.1 WT or κ-macroself mice for BM reconstitution (*Figure 7A*). For the analysis, all cells were pre-gated on the GFP⁺ population (BCL2-expressing cells) prior to gating for the individual B cell subpopulations. BCL2-overexpressing HSC from CTRL and B-KO mice reconstituted the BM of CD45.1 WT and κ-macroself transgenic mice equally well, since we obtained similar B cell frequencies and total cell numbers in all conditions (*Figure 7B and C*; *Figure 7—figure supplement 1A*). As expected, ectopic BCL2 expression increased the relative number of both κLC⁺ as well as $\lambda$ LC⁺ B cells in CTRL and Kidins220-deficient B cells in the BM of non-transgenic WT mice when compared to non-transgenic WT mice reconstituted with non-BCL2-expressing HSC (*Figure 7—figure supplement 1B, C*; *Figure 6—figure supplement 1B,C* ). Transgenic BCL2 expression in the presence of the anti-κ 'auto-antigen' did not override the induction of central tolerance, in line with previous reports (*Ait-Azzouzene et al., 2005*; *Vela et al., 2008*; *Figure 7D*; *Figure 7—figure supplement 1B*). Instead, it led to a higher percentage and total cell number of $\lambda$ LC⁺ B cells in κ-macroself transgenic mice reconstituted with HSC from CTRLs (*Figure 7E*; *Figure 7—figure supplement 1C*). In contrast, neither the percentage nor the total number of $\lambda$ LC⁺ B cells in κ-macroself transgenic mice reconstituted with B-KO HSC increased upon BCL2-overexpression, nor did they reach the levels observed in mice reconstituted with HSC from CTRLs (*Figure 7E*; *Figure 7—figure supplement 1C*). The presence of the κ-macroself antigen only

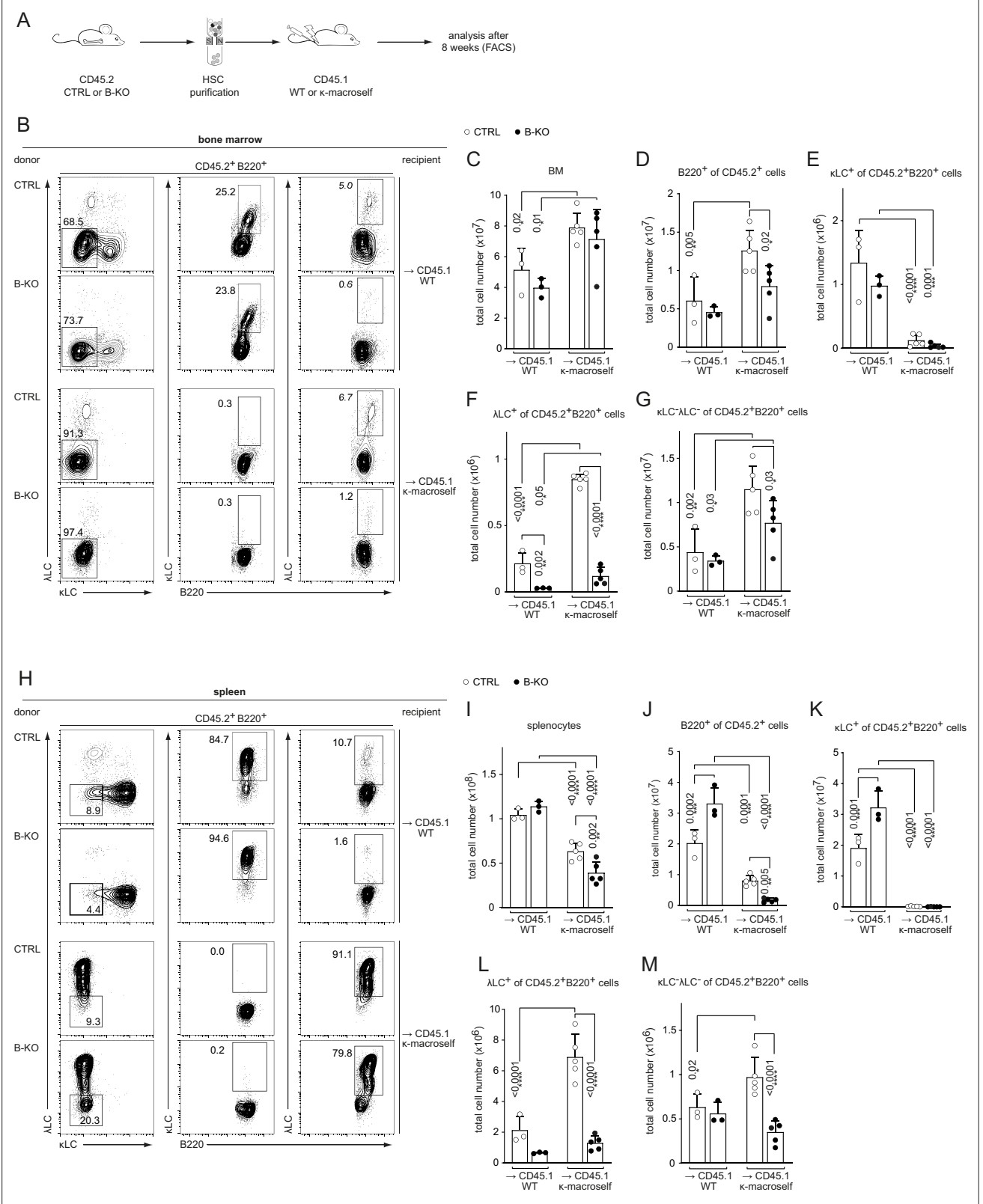

**Figure 6.** Kidins220 is dispensable for the elimination of autoreactive BCRs but necessary for the expression of innocuous $\lambda$ LC during tolerance induction. (**A**) Experimental setup for BM transfer into CD45.1 mice. HSC from CTRL and B-KO mice were isolated by negative magnetic purification. A total of 5x10⁵ cells were injected intravenously into sublethally irradiated CD45.1 WT or CD45.1 $\kappa$-macroself transgenic mice. After 8 weeks, the reconstituted mice were analyzed by flow cytometry. Representative plots of the BM (**B**) and spleen (**H**) are shown. Total cell numbers of each B cell

*Figure 6 continued on next page*

*Figure 6 continued*

compartment from BM (**C–G**) and spleen (**I–M**) are shown. For the quantification, three independent experiments were pooled (n=6–13 mice per condition). Each symbol represents one mouse. In all graphs the mean + SD is plotted. Statistical analysis was performed using One-way ANOVA with Fisher's LSD test. Only p-values <0.05 are indicated.

The online version of this article includes the following source data and figure supplement(s) for figure 6:

**Source data 1.** Kidins220 is dispensable for the elimination of autoreactive BCRs but necessary for the expression of innocuous $\lambda$ LC during tolerance induction.

**Figure supplement 1.** Kidins220 is dispensable for the elimination of autoreactive BCRs but is necessary for the expression of innocuous $\lambda$ LC during tolerance induction.

**Figure supplement 1—source data 1.** Kidins220 is dispensable for the elimination of autoreactive BCRs but is necessary for the expression of innocuous $\lambda$ LC during tolerance induction.

marginally increased the presence of BM B cells that lack surface expression of LC, and therefore BCR (*Figure 7F*; *Figure 7—figure supplement 1D*). All these findings from the BM were reproduced in the spleens with a similar phenotype (*Figure 7G–K*; *Figure 7—figure supplement 1E–H*).

We next investigated the contribution of prolonged survival in all four conditions by first gating on $\kappa$LC$^+$ or $\lambda$ LC$^+$ B cells and subsequently gating on BCL2-overexpressing (GFP$^+$; striped bars) *versus* non-overexpressing cells (GFP$^-$; clear bars; *Figure 7L and M*). Kidins220-deficient $\lambda$ LC$^+$ B cells benefit more from the overexpression of BCL2 compared to CTRL counterparts in the BM of CD45.1 WT mice. This difference however disappeared in the BM of $\kappa$-macroself transgenic mice (*Figure 7L*). The same tendencies were observed in the spleens of the respective mice (*Figure 7M*). These data revealed that the anti-apoptotic function of BCL2 has a greater impact on the generation of $\lambda$ LC B cells than of $\kappa$LC B cells. Taken together, these data support a model in which Kidins220 is dispensable for the transmission of strong BCR-mediated auto-antigenic signals during tolerance induction. However, Kidins220 is needed to rearrange the genes of the *Igl* locus and thus fulfil the repertoire with innocuous BCRs. Prolonged survival by transgenic BCL2 overexpression alone, or in combination with enforced receptor editing, failed to fully rescue $\lambda$ LC-expression.

## Kidins220 is required for optimal pre-BCR signaling

*LC* locus opening and successful recombination depends on optimal pre-BCR and/or BCR signaling. We previously demonstrated that Kidins220 is important for BCR-mediated downstream signaling of *ex vivo* stimulated primary splenic B cells (*Fiala et al., 2015*). Further, mice deficient for pre-BCR downstream signaling components, such as BTK, SPL65, or PLCγ2, show an impaired upregulation of surface markers like CD25, and major histocompatibility complex (MHC) class II concomitant with defective downregulation of CD43 (*Kersseboom et al., 2003*; *Kersseboom et al., 2006*; *Middendorp et al., 2002*; *Stadhouders et al., 2014*; *Xu et al., 2007*). Thus, we analyzed the surface expression of these markers on BM B cells from CTRL and B-KO mice. Indeed, developing B cells of B-KO mice showed a reduced percentage of CD25$^+$ cells with lower levels of CD25 on the B cell surface (*Figure 8A*). Furthermore, B-KO mice pre-B cells failed to efficiently upregulate MHCII (*Figure 8A*), whereas the relative amount of CD43-expressing B cells was increased (*Figure 8A*). In all, these data indicate reduced pre-BCR signaling. For deeper investigation, we analyzed the basal phosphorylation state of several prominent BCR signaling molecules throughout B cell development. Our data revealed an unaltered phosphorylation pattern of the proximal (pre-) BCR signaling molecules SYK and SLP65 in B-KO cells compared to CTRL B cells (*Figure 8B*). No major differences were found for the phosphorylation of p65 and IκB which are components of the NF-κB signaling pathway (*Figure 8C*). In contrast, the proportion of phospho-ERK positive pre- and immature B cells in Kidin220-deficient mice was significantly reduced when compared to CTRLs (*Figure 8D*). Next, we analyzed the basal level of reactive oxygen species (ROS) in CTRL and B-KO mice. ROS serve as important second messengers (reviewed in *Tsubata, 2020*). To this end, we isolated primary BM cells and splenocytes from B-KO and CTRL mice and labeled them with the fluorogenic dye DCFDA, in combination with a surface marker to analyze cellular ROS production within the different stages of B cell development. B cells from B-KO mice showed a significantly reduced level of ROS during almost all BM developmental stages except for large pre-B cells (*Figure 8E*). We confirmed a significant drop of ROS from the large to small pre-B cells, as previously reported (*Stein et al., 2017*). This drop was much more pronounced

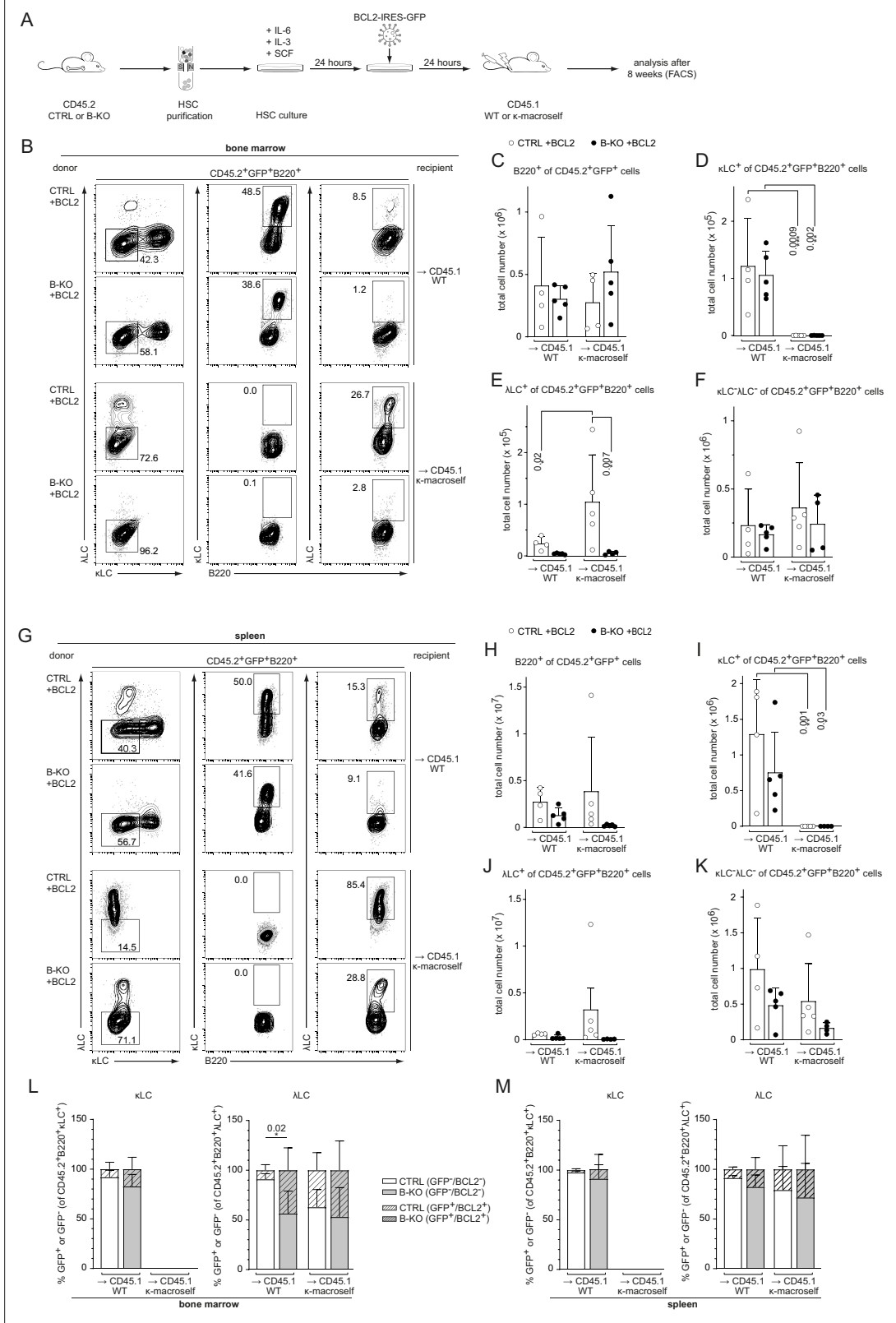

**Figure 7.** BCL2 overexpression fails to rescue $\lambda$ LC expression during tolerance induction. (**A**) Experimental setup for BM transfer of virally transduced HSC into CD45.1 WT or CD45.1 $\kappa$-macroself transgenic mice. HSC from CTRL and B-KO mice were isolated by negative magnetic purification and cultured for 24 hr in the presence of IL-6, IL-3, and SCF. The cells were then virally transduced with a BCL2-IRES-GFP overexpression plasmid and cultured for another 24 hr. A total of 5x10⁵ cells were injected intravenously into sublethally irradiated CD45.1 WT or CD45.1 $\kappa$-macroself transgenic

*Figure 7 continued on next page*

*Figure 7 continued*

mice for BM reconstitution. After 8 weeks, the reconstituted mice were analyzed by flow cytometry. (**B–K**) BCL2-expressing donor B cells were analyzed by pre-gating on GFP$^+$ cells. Representative plots of BM (**B**) and spleen (**G**) are shown. Total cell numbers of each B cell compartment from BM (**C–F**) and spleen (**H–K**) are shown. (**L, M**) Percent of donor B cells expressing BCL2 (GFP$^+$) and not expressing BCL2 (GFP$^-$) upon gating for $\kappa$ LC$^+$ (left) or $\lambda$ LC$^+$ (right) cells in the BM (**L**) and the spleen (**M**). Significance was determined by comparing the percentage of the GFP$^+$ (striped) bars between the conditions. All panels are generated from two independent experiments (n=4–9 mice per condition). Each symbol represents one mouse. In all graphs, the mean + SD is plotted. Statistical analysis was performed using One-way ANOVA with Fisher's LSD test. Only p-values <0.05 are indicated.

The online version of this article includes the following source data and figure supplement(s) for figure 7:

**Source data 1.** BCL2 overexpression fails to rescue $\lambda$ LC expression during tolerance induction.

**Figure supplement 1.** BCL2 overexpression partially rescues $\lambda$ LC expression during tolerance induction.

**Figure supplement 1—source data 1.** BCL2 overexpression fails to rescue $\lambda$ LC expression during tolerance induction.

in Kidins220-deficient developing B cells. ROS were maintained at very low levels throughout B cell development in the BM of Kidins220-deficient mice. Splenic naïve B cells of Kidins220-sufficient and -deficient mice showed similar low ROS levels. One of the main production sites of intracellular ROS are the mitochondria (***Bae et al., 2011***). Interestingly, previous reports pointed toward a connection between Kidins220 and mitochondrial function in neuronal cells (***Duffy et al., 2011***; ***Jaudon et al., 2021***). Hence, we examined the mitochondrial mass and activity in developing B cells using MitoTracker Red. However, we did not detect differences between CTRL and B-KO mice, suggesting a rather normal mitochondrial function (***Figure 8—figure supplement 1***). These results might indicate that Kidins220-deficient B cells produce less ROS due to impaired pre-BCR and BCR-mediated signaling. Reduced pre-BCR and BCR signaling might in turn prevent the opening and recombination of the *Igλ* locus in developing B cells.

In an attend to rescue the signaling defect in Kidins220-deficient B cells, we retrovirally transduced our pro-/pre-B cell cultures to overexpress BTK (***Figure 9***). BTK has previously shown to be pivotal for LC expression (***Dingjan et al., 2001***; ***Kersseboom et al., 2006***; ***Xu et al., 2007***). At day 3 after IL-7 withdrawal, BTK overexpression slightly increased the relative amount of κLC$^+$ B cells, both in CTRL and B-KO cultures. In the presence of Kidins220, the proportions of $\lambda$ LC$^+$ cells were stronger increased than those of κLC$^+$ B cells. These observation are in line with published reports suggesting a higher dependency of $\lambda$ LC B cell development on functional (pre-) BCR-mediated signaling (***Dingjan et al., 2001***; ***Kersseboom et al., 2006***). In contrast, BTK overexpression in B-KO cells failed to rescue the amount of $\lambda$ LC B cells to CTRL levels. These findings again emphasize the double function of Kidins220 in B cell development by regulating (pre-) BCR signaling and by supporting B cell survival.

## Discussion

We have previously discovered that the transmembrane protein Kidins220 binds to the BCR and regulates BCR signaling (***Fiala et al., 2015***). Deleting Kidins220 in B cells (B-KO mice) results in a severe reduction of $\lambda$ LC B cells despite normal generation of κLC B cells, generating thus a skewed antibody repertoire. Our data indicate that the absence of Kidins220 specifically prevents opening and/or transcription of the *Igl* locus. Remarkably, Kidins220-deficient B cells fail to open and recombine the *Igl* locus, even in genetic scenarios where the *Igk* genes cannot be rearranged (κ-KO). Our single-cell analysis of the frequencies of *Igl-V*-gene recombination has confirmed a clear preference for the recombination of *Iglv1*-gene in WT mice (***Boudinot et al., 1994***; ***Sanchez et al., 1996***). This also holds true for B-KO mice. The current model proposes that this preference results from the cooperative function of the $\lambda$ 1-3 and $\lambda$ 2-4 enhancers, which surround the *Igl-V*-gene segments 3' and 5' of the $\lambda$ 1 and $\lambda$ 3 family and are bound by the same transcription factors (***Collins and Watson, 2018***; ***Haque et al., 2013***). The distance of the $\lambda$ 2-4 enhancer to the $\lambda$ 1-3 enhancer might preclude that *Igl* gene recombination 5' of the $\lambda$ 2-4 enhancer profit from transcription factors and co-factors of the recombination machinery bound to the $\lambda$ 1-3 enhancer. The rearrangement of the *Iglv1* to *Iglj1c1*-gene segment eliminates the intervening sequence comprising the *Iglj3c3*-gene segment, resulting in the accumulation of $\lambda$ 1 LC B cells. Interestingly, we observed a relative increase in the usage of the genes of the $\lambda$ 3-family compared to the $\lambda$ 2-family in B-KO *versus* CTRL mice. This observation suggests that the recombination of the *Igl-V*-gene segments that are 5' of the $\lambda$ 2-4 enhancer is more depended on Kidins220 expression than the others. A possible explanation for this observation is that

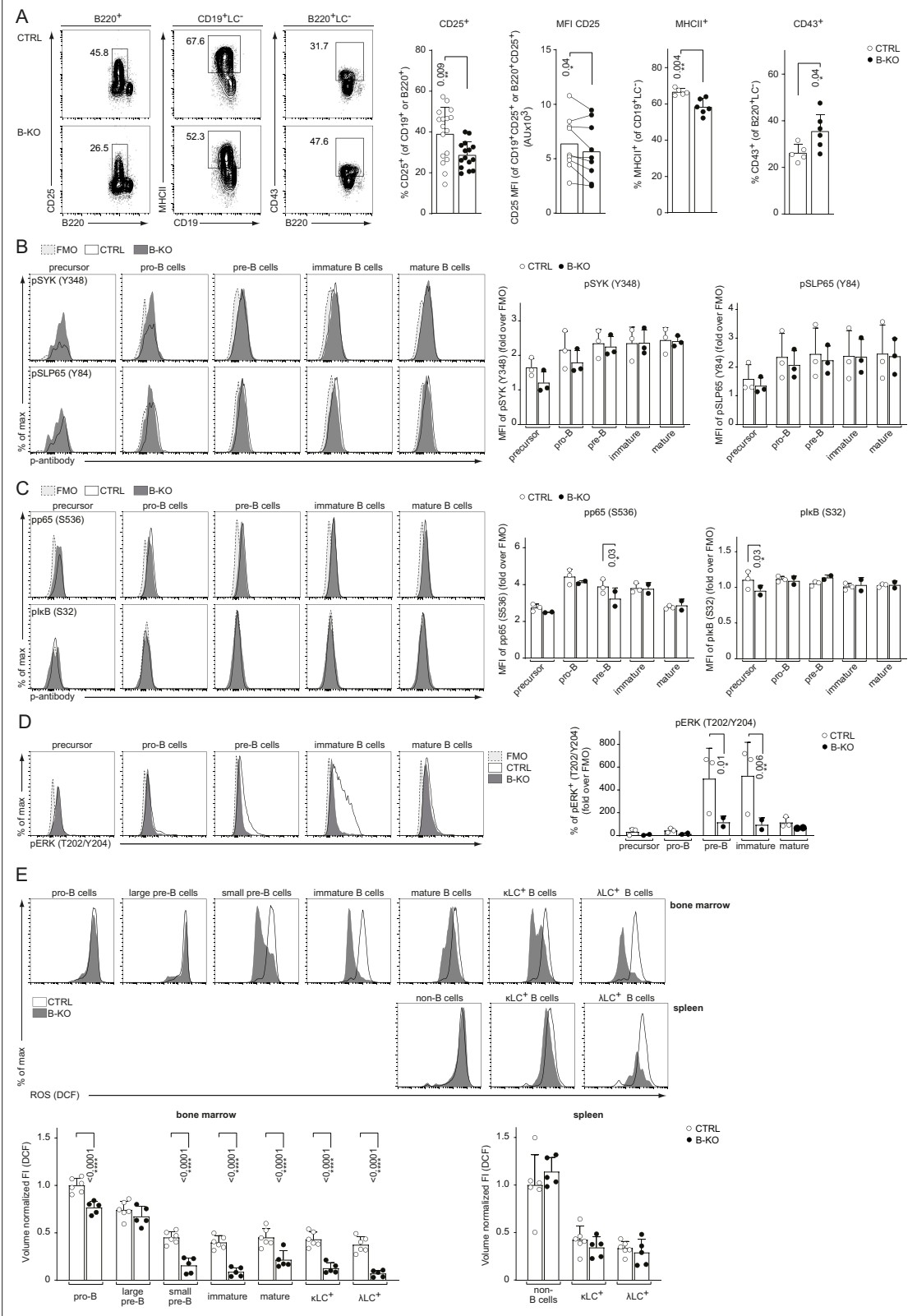

**Figure 8.** Rewired pre-BCR signaling in the absence of Kidins220. (**A**) Representative flow cytometry plots of the BM of CTRL and B-KO mice showing the surface expression of CD25, MHCII and CD43 (left). For CD25, quantification of nine independent experiments with n=15–17 mice per genotype is shown. Each dot represents an individual mouse (percentage) or the average of each group within an individual experiment (MFI). For MHCII and CD43, quantification of three independent experiments with n=5–6 mice per genotype is shown. Each dot represents an individual mouse. Statistical analysis

*Figure 8 continued on next page*

*Figure 8 continued*

was performed using paired (CD25 MFI) or unpaired Student's *t*-test. (**B–D**) BM cells were isolated and directly fixed for basal phospho-flow analysis. Representative histograms (left) and quantifications (right) of the relative MFI signal or percent of cells over the FMO are shown. (**B**) pSYK (S348) and pSLP65 (Y84). (**C**) NF-$\kappa$B signaling pathway: pp65 (S536) and pI$\kappa$B (S32). (**D**) pERK (T202/Y204). One representative experiment (out of 1–3 independent experiments) with each n=2–3 mice per genotype is shown. (**E**) B cell subpopulations from BM and spleen of CTRL and B-KO mice were analyzed by flow cytometry using specific antibodies against B220, CD117 (c-kit), CD25, $\kappa$ LC, and $\lambda$ LC. Cells were additionally stained with DCFDA to assess ROS levels. Representative histograms are shown on the top. For quantification (bottom), fluorescent intensities of the indicated metabolic marker were first normalized to the mean cell volume of each subpopulation and then normalized to pro-B cells (BM) or non-B cells (spleen). Three independent experiments were pooled; n=5–6 mice per genotype. (**B–E**) Each symbol represents one mouse. Statistical analysis was performed using Two-way ANOVA with Fisher's LSD test. In all graphs, the mean + SD is plotted. Only p-values <0.05 are indicated.

The online version of this article includes the following source data and figure supplement(s) for figure 8:

**Source data 1.** Rewired pre-BCR signaling in the absence of Kidins220.

**Figure supplement 1.** Kidins220 does not overall alter the mitochondrial function of B cells.

**Figure supplement 1—source data 1.** Kidins220 does not overall alter the mitochondrial function of B cells.

pre-BCR and BCR signaling are reduced in the absence of Kidins220 making the recombination of $\lambda$ 2-family *V*-genes unlikely. Instead, *Iglv1* is recombined to *Iglj3c3* more often in Kidins220-deficient mice. Together, our single-cell sequencing analysis supports that Kidins220 is dispensable for *Igk*, but essential for efficient *Igl* locus opening and gene recombination.

Reduced production of $\lambda$ LC B cells has been previously described in studies using mouse models with genetic deletion of signaling molecules downstream of the pre-BCR and BCR like BTK, SLP65, and PLC$\gamma$2 (*Bai et al., 2007*; *Dingjan et al., 2001*; *Hayashi et al., 2004*; *Kersseboom et al., 2006*; *Middendorp et al., 2002*; *Stadhouders et al., 2014*; *Xu et al., 2007*). In contrast, the combined deletion of some of these signaling components almost completely abolishes both *Igl* and *Igk* germ-line transcription, *LC V*-gene recombination, and protein expression (*Kersseboom et al., 2006*; *Stadhouders et al., 2014*; *Xu et al., 2007*). These data led to the interpretation that the generation of $\lambda$ LC B cells is more sensitive to defects in the signals transmitted by the pre-BCR and/or BCR (*Stadhouders et al., 2014*). Kidins220-deficiency has been previously described to dampen signaling by receptors in neurons, glial cells, adipocytes, T and B cells (*Arévalo et al., 2004*; *Deswal et al., 2013*; *Fiala et al., 2015*; *Jaudon et al., 2020*; *Jaudon et al., 2021*; *López-Menéndez et al., 2009*; *Zhang et al., 2021*). Indeed, BCR stimulation in a Kidins220 knockdown mature B cell line resulted in dampened activation of the RAS-ERK pathway due to an inefficient coupling of the BCR to the downstream Raf kinases B-Raf and Raf-1 (*Fiala et al., 2015*). Likewise, IgM-BCR-mediated stimulation of primary splenocytes from B-KO mice resulted in less phosphorylation of ERK and PLC$\gamma$2, as well as reduced calcium influx compared to CTRL cells; further highlighting the role of Kidins220 in BCR signaling (*Fiala et al., 2015*). Our data herein suggest that pre-BCR signaling is

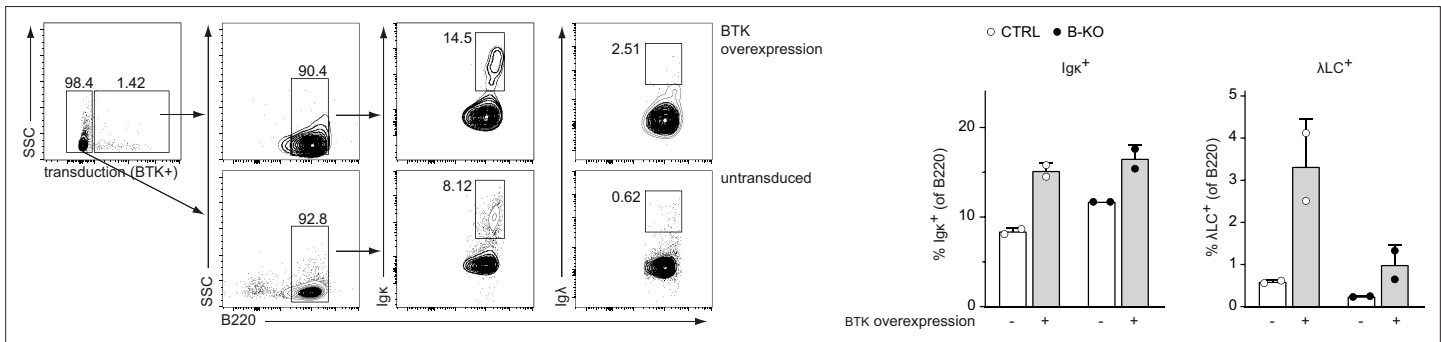

**Figure 9.** Overexpressing BTK partially rescues $\lambda$ LC development. Pro-/pre-B cell cultures were retrovirally transduced with an overexpression plasmid coding for BTK at day 7 of IL-7 culture. Two days later, IL-7 withdrawal was performed. Cells transduced with the plasmid (BTK-overexpressing cells) and untransduced cells were further analyzed for B220, $\kappa$ LC, and $\lambda$ LC expression at day 3 after IL-7 withdrawal. Gating strategy (left) and statistical analysis (right) are show. Each symbol represents one mouse, n=2 mice per genotype. In all graphs, the mean + SD is plotted.

The online version of this article includes the following source data for figure 9:

**Source data 1.** Overexpressing BTK partially rescues $\lambda$ LC development.

also reduced in the absence of Kidins220. Firstly, B-KO mice showed a reduction of the amount of MHCII[+] and CD25[+] cells as well as reduced CD25 surface expression on B cells in the BM. CD25 is upregulated upon pre-BCR signaling (*Lee et al., 2015*; *Rolink et al., 1994*), and its expression level is reduced in mice with genetic ablations of BTK, SLP65, or PLCγ2 (*Kersseboom et al., 2003*; *Kersseboom et al., 2006*; *Middendorp et al., 2002*; *Stadhouders et al., 2014*; *Xu et al., 2007*). Secondly, phosphorylation of the downstream MAPK family member ERK was reduced in pre- and immature B cell stages. Interestingly, in both of these stages active signaling takes place either ligand independent (pre-BCR) or ligand dependent (autoreactive BCR), respectively. It has been already described that (pre-) BCR signaling controls *Rag* expression via the MAPK/ERK pathway (*Mandal et al., 2009*; *Mazari et al., 2007*; *Shaw et al., 1999*). In the absence of Kidins220, the coupling of (pre-) BCR with the MAPK/ERK signaling pathway is defective resulting in insufficient upregulation of proteins needed for LC rearrangement (RAG, E47). It has been previously shown that interference with RAS-MEK signaling leads to upregulation of the negative regulator of E47, namely ID3, and a downregulation of *Tcf3 (E47)* transcripts and protein levels thus prohibiting its transcriptional function for LC rearrangement (*Mandal et al., 2009*). A similar scenario is conceivable in the context of Kidins220-deficiency. It is worth noting that rearrangement at the *Igl* locus demands higher levels of E2A activity (*Beck et al., 2009*). This further reinforces the notion that in the absence of Kidins220, the levels of RAG and E2A proteins are insufficient for rearranging the *Igl* locus while the *Igk* is rearranged normally. Thirdly, Kidins220-deficient B cells showed reduced ROS levels during BM development, especially in those stages where pre-BCR (small pre-B cells) and IgM-BCR signaling (immature B cells) is crucial. ROS are important second messengers (reviewed in *Tsubata, 2020*). In mature B cells, BCR activation induces ROS production to fully activate NF-κB signaling (*Feng et al., 2019*; *Wheeler and Defranco, 2012*). Considering the parallels between BCR and pre-BCR signaling, ROS might exert a similar role downstream of the pre-BCR. Indeed, previous reports have demonstrated that $\lambda$ LC B cell development but not κLC editing depends on NF-κB signals in line with our results (*Derudder et al., 2009*). In addition, ROS and MAPK/ERK pathways are manifold interconnected (reviewed in *Corcoran and Cotter, 2013*). ROS directly leads to the oxidization of certain amino acids in ERK leading to its activation and nuclear translocation (*Galli et al., 2008*). Furthermore, ROS are well-recognized as inactivators of phosphatases of the MAPK pathway (*Kamata et al., 2005*; *Son et al., 2011*) but also of phosphatases regulating BCR signaling (*Reth, 2002*; *Singh et al., 2005*) in non-hematopoietic cells. Thus, reduced generation of ROS in the absence of Kidins220 might lead to a higher activity of ERK-directed phosphatases, which eventually leads to the observed reduced phosphorylation of ERK in B-KO mice. Taken together, our findings support a mechanistic model in which Kidins220 mediates its effects by enhancing pre-BCR and BCR signaling. Still, we cannot formally rule out that Kidins220 might possess direct, signaling-dependent or -independent effects on the *Igl* locus opening.

Our data suggest that Kidins220 contributes to pre-B cell survival, which is necessary for *LC* gene recombination. In fact, a prolonged life-span in pre-B cells, induced by overexpression of anti-apoptotic BCL2, increases the amount of $\lambda$ LC[+] B cells (*Ait-Azzouzene et al., 2005*; *Derudder et al., 2009*; *Dingjan et al., 2001*). BCL2 overexpression rescues the generation of $\lambda$ LC B cells in scenarios in which NF-κB signaling was abolished during B cell development (*Derudder et al., 2009*). A connection between (pre-) BCR signaling and survival is well recognized (*Kraus et al., 2004*; *Meffre and Nussenzweig, 2002*; *Yasuda et al., 2008*). However, there is limited data on the exact mechanism. In particular, pre-BCR signaling might lead to the activation of the NF-κB pathway, that in turn regulates the transcription of *Pim2*, a pro-survival protein in pre-B cells (*Bednarski et al., 2012*; *Derudder et al., 2009*; *Siebenlist et al., 2005*). Pre-BCR-mediated ERK activation might induce phosphorylation of the pro-apoptotic protein BIM and thereby interfere with its inhibitory interaction with BCL2, resulting in enhanced pre-B cell survival similar to reports on IgM-BCR signaling (*Clark et al., 2014*; *Gold, 2008*; *O'Reilly et al., 2009*). In addition, pre-BCR-mediated ERK signaling might induce *Bcl2* transcription itself, thereby prolonging B cell survival during B cell development (*Gold, 2008*; *Yasuda et al., 2008*). Thus, the afore-discussed defects in coupling pre-BCR and/or BCR signaling to ERK activation in Kidins220-deficient B cells could lead to BCL2 levels that are insufficient to protect developing B cells from cell death. Indeed, *ex vivo* cultured B cell precursor from B-KO mice die faster upon IL-7 withdrawal (*Fiala et al., 2015*), and we also detected increased amounts of apoptotic cells during BM B cell development in these mice. However, overexpression of BCL2 did not fully rescue

$\lambda$ LC-deficiency in B-KO mice, suggesting that prolong survival of B cell precursors is required, but not sufficient, to optimally generate $\lambda$ LC B cells in the absence of Kidins220 expression.

The $\kappa$LC to $\lambda$ LC ratio is also affected by receptor editing. Receptor editing can lead to further recombination on either the same *Igk* allele, the second *Igk* allele or can proceed to the *Igl* locus (*Luning Prak et al., 2011*). The $\kappa$ and $\lambda$ LCs differ in the physicochemical and structural properties of their CDR3 regions, and $\lambda$ LCs are proposed to be more effective in rescuing BCRs that show autoreactivity (*Townsend et al., 2016*; *Wardemann et al., 2004*). In line with previous studies, we show here that genetically forcing receptor editing using $\kappa$-deleting ($\kappa$-macroself) mice increased the production of B cells using the $\lambda$ LC in Kidins220 competent mice (*Ait-Azzouzene et al., 2005*; *Beck et al., 2009*). In the absence of Kidins220, strong autoreactive ligands efficiently induce the downregulation of surface $\kappa$LC-BCRs indicating successful tolerance induction. However, the replacement of the autoreactive $\kappa$LC by an innocuous $\lambda$ LC was severely impaired in the absence of Kidins220. These data underline the importance of Kidins220 for *Igl* gene recombination, even under the high selective pressure imposed by the $\kappa$-macroself mice. Indeed, overexpression of BCL2 did not profoundly enhance $\lambda$ LC expression by B-KO cells in the $\kappa$-macroself recipients, highlighting again that increasing survival is not sufficient to optimally obtain $\lambda$ LC B cells in the absence of Kidins220 expression. The data obtained in the $\kappa$-macroself mice are in line with our single-cell repertoire analysis showing that we did not find any signs of potentially autoreactive BCRs in B-KO mice. Since Kidins220-deficient B cells do not open the *Igl* locus efficiently, secondary rearrangement at the *Igk* locus should be increased. Indeed, we observed a slight increase in the *Igkj5*-gene usage in B-KO mice compared to CTRLs. *Igkj5* is associated with several rounds of secondary rearrangement and often represents the last rearrangement before silencing of the *Igk* locus by recombining sequence recombination (*Prak et al., 1994*; *Retter and Nemazee, 1998*).

To conclude, our study demonstrates that Kidins220 supports pre-B cell survival and promotes pre-BCR and BCR signaling via ROS and ERK to allow *Igl* locus opening and $\lambda$ LC expression during both homeostatic B cell development and receptor editing. The heightened sensitivity of the *Igl* locus, compared to the *Igk* locus, to reductions in expression and activity of recombination machinery proteins regulated by pre-BCR and BCR signals (*Bai et al., 2007*; *Beck et al., 2009*; *Johnson et al., 2008*; *Novak et al., 2010*; *Pathak et al., 2008*; *Vela et al., 2008*; *Verkoczy et al., 2005*), explains why the absence of Kidins220 specifically impacts *Igl* gene rearrangement. This study adds thus to our understanding of the underlying mechanism regulating the differential expression of the $\kappa$ and $\lambda$ LCs in B cells and the generation of a self-tolerant repertoire.

## Materials and methods
### Cells and mice

Primary murine BM or pro-/pre-B cells were cultured in Opti-MEM containing 10% fetal calf serum (FCS), 2 mM L-Glutamine, 50 U/ml penicillin, 50 µg/ml streptomycin and 50 µM β-mercaptoethanol in a humidified saturated atmosphere at 37 °C with 5% $CO_2$. If needed, medium was supplemented with 5 ng/ml IL-7 (Peprotech). Primary murine HSC were cultured in complete DMEM GlutaMAX supplemented with 20% FCS, 50 U/ml penicillin, 50 µg/ml streptomycin, 50 µM β-mercaptoethanol and IL-3 (20 ng/ml, Peprotech), IL-6 (50 ng/ml, Peprotech) and SCF (50 ng/ml, Peprotech) in a humidified saturated atmosphere at 37 °C with 7.5% $CO_2$.

For viral transduction, human embryonic kidney (HEK) 293T cells and Phoenix-eco cells were used to produce virus-containing supernatants. They were cultured in complete DMEM GlutaMAX medium supplemented with 10% FCS, 10 mM HEPES, 10 µM sodium pyruvate, 50 U/ml penicillin and 50 µg/ml streptomycin in a humidified saturated atmosphere at 37 °C with 7.5% $CO_2$. Both cell lines were obtained from ATCC (#CRL-11268 and #CRL-3214) and were tested negatively for mycoplasma.

The Kidins220mb1hCre (*Fiala et al., 2015*), iEκT (*Takeda et al., 1993*), vav-BCL2$^{Tg}$ (*Ogilvy et al., 1999*), C57BL/6-Ly5.1 (CD45.1 WT) and CD45.1 κ-macroself mice (*Ait-Azzouzene et al., 2005*) were bred under specific pathogen-free conditions. All mice were backcrossed to C57BL/6 background for at least 10 generations. The following mice were used: Kidins220 locus: CTRL mice carried one wild-type allele (+) and one floxed (fl) or deleted (-) allele as well as B-KO mice carried either two fl alleles or one fl and one deleted (-) allele; iEκT locus: animals carrying the neomycin resistance cassette within the endogenous intronic κ enhancer (iEκT) on both alleles served as κ-KO mice. Control animals

were heterozygous for iEκT expression; vav-BCL2$^{Tg}$ mice: animals were generated by crossing non-transgenic mice with BCL2-transgenic mice. All mice used in our experiments expressed the mb1Cre heterozygously. Mice carrying the κ-macroself transgene were bred by crossing non-transgenic mice with κ-macroself transgenic mice. Mice were sex and age matched whenever possible and analyzed between 8 and 20 weeks of age. All animal protocols (G12/64) were performed according to German animal protection laws with permission from the responsible local authorities. The sample size was determinate assuming a effect size of 1.06, power of 80% and a two-sided significance level of 5%.

## Flow cytometry and cell sorting

Single cells suspensions were gained from BM and spleens. Erythrocytes were removed prior to flow cytometry analysis by incubation in erythrocyte lysis buffer containing 150 mM $NH_4Cl$ and 10 mM $KHCO_3$ for 2 (BM) or 4 (spleen) minutes at room temperature. 0.3–1×10$^6$ cells were stained in PBS, containing 2% FCS and the respective antibodies for 20 min on ice. Cells were washed and measurements were performed using a Gallios (Beckman Coulter) or Attune NxT (Thermo Fisher Scientific) flow cytometer. Unspecific antibody binding was prevented by preincubation with TruStain FcX (clone 93, Biolegend).

Primary murine immature B cells for BCR repertoire analysis were sorted on a MoFlo Astrios EQ cell sorter (Beckman Coulter) using specific antibodies against B220, IgD, and IgM (Fab Fragment). HSC were enriched by negative selection from total BM using biotinylated antibodies against CD3, B220, Ter119, CD11b, Ly6G/Ly6C followed by incubation with paramagnetic beads and magnetic cell sorting (MACS, Miltenyi Biotec).

## Antibodies

The following murine antibodies were used in flow cytometry: anti-B220(CD45R)-PECy7 (RA3-6B2) (eBioscience #25-0452-82), anti-IgM-PE (eB121-15F9) (Thermo Fisher Scientific #12-5890-82), anti-IgD-eFluor450 (11–26 c) (eBioscience #48-5993-82), anti-IgM-Fab Fragment-Alexa Fluor 647 (Jackson ImmunoResearch Laboratories, Inc #115-607-020), anti-Ig $\lambda$ 1, $\lambda$ 2, $\lambda$ 3-FITC (R26-46) (BD Biosciences #553434), anti-Ig $\lambda$ 1, $\lambda$ 2, $\lambda$ 3-bio (R26-46) (BD Biosciences #553433), anti-Igκ-V450 (187.1) (BD Biosciences #561354), anti-CD117 (c-kit)-Brilliant Violet 421 and Brilliant Violett 605 (2B8) (Biolegend #105827 and #105847), anti-CD25-APC (PC61.5) (Thermo Fisher Scientific #17-0251-82), anti-CD45.2-PE (104) (BD Biosciences #560695), anti-CD45.1-APC (A20) (eBiosciences #17-0453-82), anti-CD3-biotin (145–2 C11) (eBioscience #13-0031-82), anti-B220(CD45R)-biotin (RA3-6B2) (eBioscience #13-0452-82), anti-Ter119-biotin (TER-119) (eBioscience #13-5921-82), anti-Ly6G/Ly6C (RB6-8C5) (eBioscience #14-5931-82), anti-CD11b/Mac1 (M1/70) (eBioscience #13-0112-82), anti-CD19-PB (6D5) (Biolegend #115523), anti-MHCII-biotin (M5/114.15.2) (eBioscience #13-5321-82), anti-CD43-PE (S7) (BD Biosciences #553271), anti-IL-7Ra (CD127)-PE (A7R34) (eBioscience #12-1271-82), anti-pAKT (S473) (D9E) (cell signaling #4060), anti-pFOXO1 (S256) (cell signaling #9461), anti-pSYK (Y348)-PE (BD Phosflow #558529), anti-pSLP65 (Y84)-PE (BD Phosflow #558442), anti-pERK1/2 (T202/Y204) (cell signaling #9101), anti-pIkB (S32) (14D4) (cell signaling #2859), anti-pp65 (S536) (93H1) (cell signaling #3033), secondary goat anti-Rabbit IgG (H+L) Alexa Fluor 647 (Invitrogen #A-21245). FITC Annexin V Apoptosis Detection Kit I was purchased from BD Biosciences (#556547) and used according to manufacturer's instructions.

## Phospho-flow

Single cells suspensions from BM in PBS were immediately fixed with the same volume of 4% PFA for 10 min at RT. After centrifugation, cells were incubated again for in 4% PFA for 15 min at RT. Cells were washed twice in PBS containing 2% FCS. Subsequently, cells were permeabilized with ice cold methanol (87.7%) for 30 min on ice and then washed twice with PBS containing 2% FCS. Next, cells were stained as usual with antibodies to distinguish the developmental stages in the BM. After washing, cells were stained with the respective phospho-antibody at 4 °C overnight. If needed, secondary antibody stain was done on ice for 2 hr the following day.

## Primary BM cultures

Primary BM cultures were essentially generated by isolating total BM from the femur as previously described (*Johnson et al., 2012*). Briefly, erythrocytes were removed by incubation in erythrocyte

lysis buffer (see section Flow Cytometry) for 2 min at room temperature. 3 ml of $5×10^6$ cells/ml were cultured in one well of a p6 culture dish for 7 days in the presence of 5 ng/ml IL-7 in complete Opti-MEM. Fresh medium was added after 4 days. Afterwards, fresh medium was added every 3 days and cultures were split if needed. For IL-7 removal, cells were harvested and washed at least twice with an excess of medium without IL-7. Cells were plated in a p24 culture dish at a concentration of $2×10^6$ cells/ml in complete Opti-MEM for subsequent assays.

## Viral transduction

Murine retrovirus-containing supernatants were obtained by transfecting Phoenix-eco cells using the PromoFectin (PromoKine) reagent according to manufacturer protocols using pMIG-hBCL2-IRES-GFP, pMIG-mBtk-IRES-GFP, pMIG-IRES-GFP (Mock) or GFPi-reporter plasmids. Viral supernatant was collected and filtered after 48 hr and used directly. $1.5x10^6$ pro-/pre-B cells (or HSCs) were resuspended in 1 ml of viral supernatant containing Polybrene (1 µg/ml) and IL-7 (5 ng/ml) (or in complete HSC culture medium) and spin infected by centrifugation (90 min, 2500 rpm, 30 °C). The viral supernatant was then removed, and cells were cultured under optimal conditions for subsequent assays or injection into mice 24 hr later.

Lentivirus was obtained by co-transfecting HEK 293T cells with LeGO-iG2-hBCL2-IRES-GFP, pCMVDR8.74 and pMD2G plasmids using Polyethyleneimine (PEI, Polysciences). Viral supernatant was harvested and combined 24- and 48 hr post-transfection. Lentiviral particles were enriched by overlaying in a 1:5 ratio on a 10% sucrose layer and centrifuging (4 hours, 10.000 x $g$, 8 °C). The pellet containing lentiviral particles was resuspended in DMEM GlutaMAX w/o supplements and stored at –80 °C. The viral titers were assessed by determining the multiplicity of infection (MOI). Briefly, $5x10^4$ HEK 293T cells per well were seeded in a p24 well plate in 1 ml medium. An aliquot of the concentrated virus was diluted 1:100 in medium. Various volumes (0, 1, 5, 10, 25, and 50 µl) of lentivirus dilution were added to the cells. After 48 hr GFP expression was analyzed by flow cytometry and lentivirus titer was calculated using the following formula: Transduction units per ml = (number of cells x percent GFP$^+$ cells x dilution factor) / (ml of lentivirus dilution). Primary HSC were spin infected with a MOI of 10 (90 min, 2500 rpm, 30 °C) in complete HSC culture medium 24 hr prior to injection into the mice.

## qRT-PCR

Total RNA was isolated using TRIzol reagent (Thermo Fisher Scientific) according to manufacturer's instructions. RNA concentration was assessed using Nanodrop. One µg of RNA was treated with DNase for 30 min at 37 °C prior to cDNA synthesis. cDNA was prepared with oligo dT primers according to the manufacture's protocol (Thermo Scientific). qRT-PCR was performed using Fast Start Universal SYBR Green Master (ROX) (Roche) according to manufacturer's protocol. For amplification, gene-specific primers were used with a one-step protocol with an annealing temperature of 60 °C. Expression levels were normalized to the expression of the house keeping gene *Hprt* in pro-/pre-B cell cultures and to both, *Hprt* and *Actb*, in directly sorted cells. Relative expression levels ($2^{-\Delta Ct}$) were shown.

The following primers were used (in 5' to 3' direction): *Igk$^{0.8}$* for (CAGTGAGGAGGGTTTTTGTA CAGCCAGACAG), *Igk$^{0.8}$* rev (CTCATTCCTGTTGAAGCTCTTGA), *Igl$^1$* for (CTTGAGAATAAAATGC ATGCAAG), *Igl$^1$* rev (TGATGGCGAAGACTTGGGCTGG), *Rag1* for (ACCCGATGAAATTCAACACC C), *Rag1* rev (CTGGAACTACTGGAGACTGTTCT), *Rag2* for (ACACCAAACAATGAGCTTTCCG), *Rag2* rev (CCGTATCTGGGTTCAGGGAC), *E2A* for (GGGAGGAGAAAGAGGATGA), *E12* rev (GCTCCGCC TTCTGCTCTG), *E47* rev (CCGGTCCCTCAGGTCCTTC), *Hprt* for (GTTAAGCAGTACAGCCCCAAA), *Hprt* rev (AGGGCATATCCAACAACAAACTT), *Actb* for (GGCTGTATTCCCCTCCATCG), *Actb* rev ( CCAGTTGGTAACAATGCCATGT) *Irf4* for (CTCTTCAAGGCTTGGGCATT), *Irf4* rev (TGCTCCTTTTTT GGCTCCCT).

## ROS and mitochondrial staining

Single-cell suspensions were isolated from BM and spleens. Erythrocytes were removed as described before (see section: Flow cytometry). For ROS staining, $2x10^6$ cells were incubated in Opti-MEM with $H_2$DCFDA (10 µM; Invitrogen) for 30 min at 37 °C. Mitochondria were stained by incubating $2x10^6$ cells in 500 µl Opti-MEM supplemented with MitoTracker Red CMXRos (60 nM; Invitrogen) for 30 min at

37 °C. Afterwards, cells were washed twice and stained for surface markers for flow cytometry (see section: Flow cytometry). For quantification, the fluorescence intensity of $H_2DCFDA$ was normalized to the cell volume by FSC-W as an indicator of cell diameter as previously described (*Stein et al., 2017*; *Tzur et al., 2011*).

### Data analysis

Flow cytometric data were analyzed using FlowJo V10 (Tree Star, Inc) software. Data analysis and Presentation was done with GraphPad Prism10. Prior to statistical analysis, data were tested for normality using Shapiro Wilk test.

### Antibody repertoire library preparation and sequencing

For antibody repertoire analysis, the BM of three individual mice of each genotype (CTRL and B-KO respectively) were pooled and immature B cells (B220$^+$IgM$^+$IgD$^-$) were FACS sorted using appropriate antibodies (or Fab-fragments for IgM). Single B cell antibody V(D)J libraries were prepared with the 10 X Genomics Chromium Single Cell V(D)J platform version 1, allowing immune profiling of full-length antibody variable HC and LC (10XGenomics). Briefly, samples were loaded onto the Chromium Controller and partitioned into Gel Beads-in-emulsion (GEMs) containing single cells. The mRNA was reverse transcribed into barcoded cDNA. HC and LC full V(D)J variable sequences were amplified with a two-step PCR. Sequences were then fragmented and indexed with indices for Illumina sequencing. Quality control of the materials were obtained throughout the process and library concentration quantification was performed using an Agilent Bioanalyzer. Antibody library pools were sequenced on an Illumina MiSeq instrument at 2x300 bp paired-end reads using the MiSeq Reagent kit v3 (600 Cycles).

### Annotation, preprocessing and statistical analysis of antibody repertoires

Antibody repertoire data generated through the 10 X Genomics V(D)J platform was demultiplexed using Cellranger mkfastq and subsequently annotated with IgBLAST version 1.14 and Cellranger version 4.0. Preprocessing included filtering for retaining CDR3s longer than amino acids, selection of productive sequences and retaining only CDR3s occurring more than once in the repertoire, with productive sequences defined as sequences that are in-frame and contain no stop codons. Statistical analysis of antibody repertoire datasets was conducted using R 4.0.5. The Pearson's correlation coefficients were calculated to measure the strength of the linear association between the *V*- and *J*-germline gene frequencies between CTRL and B-KO mice for HC and LC. The correlation is significant at the 0.05 level (2-tailed).

### Material availability statement

All material used in the present study are available from the corresponding author upon justified request.

## Acknowledgements

We thank K Fehrenbach for technical assistance, the group of M Erlacher, M Reth and F Cesca for providing mice and J Jellusova for technical advice with ROS staining. We further want to thank core facilities at the BIOSS, CIBSS and ZTZ in Freiburg. We thank P Nielsen, M Reth, K Schachtrup for stimulating discussions and intellectual input, and P Nielsen and S Pathan-Chhatbar for carefully reading the manuscript. This study was supported by the German Research Foundation (DFG) through SFB1160 (Project ID: 256073931 - B01 to SM), which fully supported AMS. In addition, SM and WWS are supported by the DFG through BIOSS - EXC294 and CIBSS - EXC2189 (Project ID 390939984), and SM through MI 1942/4-1 (Project ID: 501418856) and SFB1479 (Project ID: 441891347 P15). GJF was supported by the DFG through GSC-4 (Speman Graduate School).

# Additional information

## Competing interests

Enkelejda Miho: owns shares in aiNET GmbH. The other authors declare that no competing interests exist.

## Funding

| Funder | Grant reference number | Author |
| --- | --- | --- |
| Deutsche Forschungsgemeinschaft | BIOSS - EXC294 | Wolfgang WA Schamel Susana Minguet |
| Deutsche Forschungsgemeinschaft | CIBSS - EXC2189 | Wolfgang WA Schamel Susana Minguet |
| Deutsche Forschungsgemeinschaft | SFB-1160 ID:256073931 | Susana Minguet |
| Deutsche Forschungsgemeinschaft | SFB1479 (Project ID: 441891347 P15) | Susana Minguet |
| Deutsche Forschungsgemeinschaft | MI 1942/4-1 (Project ID: 501418856) | Susana Minguet |

The funders had no role in study design, data collection and interpretation, or the decision to submit the work for publication.

## Author contributions

Anna-Maria Schaffer, Conceptualization, Data curation, Formal analysis, Supervision, Validation, Investigation, Visualization, Writing – original draft, Writing – review and editing; Gina Jasmin Fiala, Conceptualization, Data curation, Formal analysis, Supervision, Investigation, Writing – review and editing; Miriam Hils, Data curation, Investigation, Visualization; Eriberto Natali, Software, Formal analysis, Visualization; Lmar Babrak, Data curation, Software, Formal analysis, Investigation, Visualization; Laurenz Alexander Herr, Data curation, Investigation; Mari Carmen Romero-Mulero, Formal analysis; Nina Cabezas-Wallscheid, Supervision, Investigation; Marta Rizzi, Conceptualization, Supervision, Investigation, Writing – review and editing; Enkelejda Miho, Conceptualization, Formal analysis, Supervision, Investigation; Wolfgang WA Schamel, Conceptualization, Supervision, Funding acquisition, Investigation, Writing – review and editing; Susana Minguet, Conceptualization, Supervision, Funding acquisition, Validation, Investigation, Visualization, Writing – original draft, Project administration, Writing – review and editing

## Author ORCIDs

Anna-Maria Schaffer ⓘ http://orcid.org/0000-0002-2561-0820
Eriberto Natali ⓘ http://orcid.org/0000-0002-9100-807X
Laurenz Alexander Herr ⓘ http://orcid.org/0000-0002-1035-2558
Wolfgang WA Schamel ⓘ http://orcid.org/0000-0003-4496-3100
Susana Minguet ⓘ http://orcid.org/0000-0001-8211-5538

## Ethics

All animal protocols (G12/64) were performed according to German animal protection laws with permission from the responsible local authorities.

## Decision letter and Author response

Decision letter https://doi.org/10.7554/eLife.83943.sa1
Author response https://doi.org/10.7554/eLife.83943.sa2

# Additional files

## Supplementary files

• MDAR checklist

## Data availability

All data generated or analysed during this study are included in the manuscript and supporting files.

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
