## [Editor Report]

This manuscript addresses an important question and provides some clear instructions on the use of the λ chain locus. In particular, it highlights the role of Kidins220 in extending B cell survival so as to increase the time available for the λ light chain locus to open for VJ recombination, it also shows that extended survival time alone is not sufficient to enhance λ light chain recombination and this represents an important contribution to our understanding of B cell repertoire. The data are convincing although some mechanistic questions remain.

---

## [Decision Letter]

[Editors' note: this paper was reviewed by Review Commons.]

---

## [Author Response]

General statement:

We thank the Reviewers for critical reading our Manuscript entitled "Kidins220 regulates the development of B cells bearing the λ light chain" by Schaffer et al. We thank them for all suggestions to improve our current manuscript. We acknowledge the patience of the reviewers and editorial team, it has taken quite some time to finalize this revision due to unforeseen issues with the animal colony. We have aimed in the revised version to provide deeper mechanistic insights to extend our current knowledge of the differential regulation of light chain expression in B cells. Briefly,

We have improved the graphical representation of our data as well as the clarity of some statements as requested.We have performed experiments to provide deeper mechanistic insights into the role of Kidins220 in pre-BCR signalling (new Figures 2D, 4E, 8A and Figure 2— figure supplement 1, Figure 4—figure supplement 1 and Figure 8—figure supplement 1).We have also included new reconstitution experiments further supporting the link between reduced pre-BCR signalling in the absence of Kidins220 and generation of λLC-bearing B cells (Figure 9).

Reviewer comments:

Reviewer #1 (Evidence, reproducibility and clarity (Required)):This is a very well performed study on the role of a transmembrane protein called Kidins220 in B cells. This protein has previously been shown by the authors to be a binding partner of the BCR. In B-cell specific KO mice (B-KO) they had previously found an almost complete lack of immunoglobulins with λ light chains. In this manuscript the authors determine the mechanism of this defect in much detail. They show a failure to recombine the genes of the λ LC locus. Kidins220 enhances the survival of pre-B cells and extends the time of the opening of the λ LC locus, to allow rearrangement. Interestingly, extending the survival of pre-B cells of B-KO mice by BCl-2 overexpression does not rescue this λ light chain defect. With an autoreactive transgenic model the authors further demonstrate that Kidins220 is not essential for tolerance induction, but even in this model where receptor editing is enforced, for a complete λ LC rearrangement. Overall the data are very convincing.I have only minor comments for changes in the presentation and arrangement of the data, which are nevertheless necessary for better clarity of the manuscript.

1. I do not understand the presentation of data in Figure 1G. The description in the legend is too little. As I am not familiar with this method of showing an antibody repertoire, I do not understand it. I guess this will be similar for other readers as well. Please explain the method better that the figure can be understood.

We thank the Reviewer for pointing this out. We are happy to improve the comprehensibility of this paragraph by better explaining how the analysis of the antibody repertoire was performed (former Figure 1G, now Figure 1I). We followed the method developed in *Miho et al., 2019*. Briefly, we have created a network in which each CDR3 clone (represented by a node) is connected to another CDR3 clone via a similarity edge if and only if, their sequence differs by one amino acid (Levenshtein distance = 1). In other words, nodes that differ by only one amino acid in their CDR3 sequence are linked by edges and appear clustered. Clustered clones are indicative of antigen-dependent clonal expansion and selection. Clones which sequences differ in more than one amino acid, are not connected in our representation.

Clusters indicating clonal expansion were seen more often in the B-KO repertoire than on the CTRL repertoire. We have now indicated the clustered clones with arrow-heads to improve clarity. We have revised this section and implemented further information to improve the comprehensibility of this paragraph.

Please find our editions in lines 169-173 of the main text and lines 1191-1194 of the Figure legend. Please note, that it was formerly Figure 1G and is now Figure 1I on the revised manuscript.

2. Figure 3, Figure S3. The is the first example of several figures with FACS data, where relative percentages of cell populations are shown in the main figure (Figure 3), but total cell numbers are shown in the supplementary figure (Figure S3). The total cell numbers are more important, because they are clearer. They should be shown in the main Figure , the relative numbers as supplementary. Please exchange C-G and I-O with the respective parts of Figure 3S. One example of the unclear data of relative cell numbers is Figure 3M. By looking at the Figure the reader thinks that their is a increasing rise of the MZ B cell numbers. However Suppl.Figure 3M shows that they are reduced in total numbers in the last mouse line on the right, just less affected than transitional or FO B cells.

We agree with this Reviewer that in many cases the graphs representing the total cell numbers are clearer and support our message better. Thus, we have exchanged the panels in the new version of our manuscript as following: total numbers are now displayed in Figures 3, 5, 6, and 7, while the percentages are shown now in the corresponding Figure supplements.

3. Figure 5, Figure S5. The same story. It is a rather complicated presentation of the relative numbers in C-G. The total cell numbers in Figure S5 are much clearer. Pease exchange!. In lines 288-290 the authors describe the mature B cells of the BM as IgM+. This is wrong. The authors did use an B220 antibody or fluorophore that does not separate IgM+ B220 low and B220 high cells very well, so I suppose they used the IgM+ IgD+ gating for mature cells in the bone marrow?

We have exchanged the Figures as suggested, please see our detailed answer to question #2.

We thank the Reviewer for noticing the mistake of saying “mature IgM^+^ B cells”. Indeed, we have used IgM and IgD to separate immature and mature B cells in the BM. We now rewrote the text in lines 337-339 accordingly in the new version. It now reads: “IgM^+^IgD^+^ mature B cells clearly accumulated in the BM of -B-KO BCL2-transgenic mice…*”*

4. Figure 6 and Fg.7. Please also use the total cell numbers here.

We exchanged the requested panels in the new version for Figures 6 and 7. Please see our detailed answer to question #2.

5. Line 372. I did not understand the sentence of "innocuous λ LC, possibly leaving holes in the BCR repertoire". Please explain!

Thanks for highlighting the lack of clarity. We have here used the concept of “holes in the BCR repertoire”, that was already put forward by Goodnow back in the 90’s. (*Goodnow, 1996*). The concept hypothesizes that the establishment of tolerance, removing self-reactive B cells from the repertoire, generates “holes in the repertoire” since the lost B cells could have also potentially recognized foreign epitopes that structurally resemble determinants on selfantigens. Thus, the “holes” created in the B cell repertoire could be exploited by microbial pathogens. Building on this concept, the absence of λLC B cells in Kidins220 B-KO mice alters the B cell repertoire generating an incomplete repertoire with “holes”, since those antigens which recognition is dominated by the λLC will not be recognized. This is not only a theoretical possibility. For example, primary immune responses driven by the hapten NP are accounted by BCRs carrying the λLC (*Reth et al., 1978*). Therefore, the BCR repertoire of B-KO mice have a “hole” lacking BCRs that are able to identify this particular antigen, as we have already demonstrated (*Fiala et al., 2015*). We have now edited the paragraph seeking to improve the understandability of our message as follows: *“Consequently, in B-KO mice, there is a deficiency of B cells capable of responding to foreign epitopes recognized by BCRs carrying the λLC, resulting in what has been described as “holes in the BCR repertoire” (Goodnow, 1996).*

Please find our editions in lines 425-427 of the main text.

6. For the pre-BCR signaling study: Why was ROS production chosen as a readout for this? Why not e.g. phospho-FACS of proximal or distal intracellular signaling proteins? That would have been more straightforward, according to my opinion.

It has been well documented that ROS are produced upon BCR engagement (*Feng et al., 2019; Tsubata, 2020;Wheeler and Defranco, 2012*). Still, we agree that additional readouts would strengthen our data. We have thus performed phospho-flow FACS analysis of signaling molecules downstream of the (pre-) BCR.

Firstly, we have tested multiple phospho-antibodies and selected those that fulfilled the following criteria: the phospho-flow signal of the respective antibody shifted towards positiveness upon anti-IgM stimulation only in those populations expressing the targeted molecules. We have thus selected anti-phospho-SYK (Y348), anti-phospho-SLP65 (Y84), antiphospho-p65 (S536), anti-phospho-IκB (S32), anti-phospho-ERK (T202/Y204), and antiphospho-AKT (S473).

Secondly, we have comparatively investigated the phosphorylation state of the selected molecules during B cell development in the BM of CTRL and B-KO mice at basal conditions. No statistically significant differences in proximal signaling molecules downstream of the (pre-) BCR were detected between CTRL and B-KO mice as shown by analyzing the activatory phosphorylation of SYK (Y348) and SLP65 (Y84) (new Figure 8B). In contrast, the proportions of cells positive for phospho-ERK (T202/Y204) were reduced in B-KO pre- B cells and immature B cells compared to CTRL (new Figures 8B and 8D). Remarkably, pre- B cells and immature B cells are indeed the maturation stages expected to integrate pre-BCR and BCR signals, respectively. These results are in line with our previous report highlighting a link between the BCR and the ERK pathway (*Fiala et al., 2015*).

We did not detect major differences with respect to NF-κB signaling between B-KO and CTRL B cells (Figure 8C). In contrast, PI3K signaling was enhanced (phospho-AKT (S473)) signaling during early B cell development in B-KO compared to CTRL mice (new Figure 4— figure supplement 1C). PI3K signaling during early B cell development is primary triggered by IL-7R (reviewed in (*Clark et al., 2014*)). Indeed, increased IL-7R signaling results in enhanced AKT phosphorylation, enhanced phosphorylation of its target protein (phosphoFOXO1 (S256)) (new Figure 4—figure supplement 1C), and consequently, the retention of FOXO1 proteins in the cytoplasm, reducing thus FOXO1 target gene expression like *Rag1/2.* In fact, *Rag2* expression is reduced in B-KO cells as shown in new Figure 4E.

Taken together, our additional data point towards a role of Kidins220 in supporting (pre-) BCR signaling via ROS and ERK, as well as IL-7R signaling via AKT/FOXO1. The absence of Kidins220 notably affect developmental stages in which the B cells are highly dependent on signals supporting survival and V(D)J recombination. Hence, λLC B cells are the most affected by the absence of Kidins220.

The new data are displayed in new Figure 8B-D and new Figure 4—figure supplement 1B and C. In addition, further experiments supporting the role of Kidins220 coupling signaling of the pre-BCR and BCR to the development of λLC B cells are provided in new Figure 9. For a detailed explanation of those experiments, please refer to question #2 of Reviewer #2.

Our editions are in the main text lines 300-307 and 482-489.

Reviewer #1 (Significance (Required)):It is a significant and important study about the regulation of antibody light chain rearrangement by a B cell signaling protein. Although the research question was quite specific, the findings are of greater relevance for regulation of Ig light chain rearrangement. This is important for the generation of the full antibody repertoire, but also for central tolerance induction in the BM.

We thank the Reviewer for his/her thoughtful synthesis and for highlighting the significance of our work.

Reviewer #2 (Evidence, reproducibility and clarity (Required)):In their manuscript "Kidins220 regulates the development of B cells bearing the λ light chain" Schaffer et al. describes the important role of Kidins220 in the development of λ light chain (LC)+ B cells supposedly by regulation of the opening up of the λ LC genetic locus. Λ LC is important for receptor editing and for the rescue of the developing B cells with autoreactive BCR. The manuscript presents a tour de force in terms of applying various combinations of mouse models including strain crossings and bone marrow chimeras to assess the effects of B cell survival and tolerance. Interestingly, the generation of λ-LC B cells cannot be efficiently induced even when Kidins220-KO is combined with a kappaLC-KO model or when the receptor editing is forced by kappamacroself mice. The experiments are carried out and reported carefully, and the data is robust. The phenotype is interesting and has implications in the BCR repertoire. Mechanistically, the authors show reduced pre-BCR signalling and suggest that as the underlying reason of λ-LC defect.

1. The group showed in their earlier paper the role of Kidins220 in BCR signalling and B cell activation including antibody responses (ref 37). Here, reduced signalling outcome of pre-BCR signalling is reported. However, this aspect remains somewhat superficial and, notably, no experimental data is shown that would convincingly link the reduced signalling to the defects in the λ locus. Could this hypothesis be experimentally explored?

We acknowledge that our manuscript will benefit from more experiments exploring the role of Kidins220 in pre-BCR signaling and the rearrangement of the *λLC* locus. This suggestion is in line with the opinion of Reviewer#1. We have performed phospho-flow FACS analysis of signaling molecules downstream of the (pre-) BCR (new Figure 8B-D) and reconstitution experiments by overexpressing BTK in pro-/pre-B cells (new Figure 9). We would like to refer you to question #6 of Reviewer#1 for a detailed explanation of new Figure 8B-D, and to the next question (question #2, Reviewer #2) for a detailed explanation of new Figure 9.

We are confident that the new experiments further support our hypothesis that Kidins220 plays a role in pre-BCR signaling.

Our editions are in the main text lines 300-307, 482-489 and 509-520.

2. Could the phenotype be rescued by enhancement of the pre-BCR signaling?

We thank the Reviewer for suggesting rescue experiments.

We have used our established IL-7 BM cultures to overexpress wildtype BTK in pro-/preB cells aiming to rescue the generation of λLC B cells by enhancing pre-BCR signaling.

We have then compared the generation of λLC B cells between CTRL and B-KO cells upon IL-7 withdrawal. As shown in the new Figure 9, the proportion of λLC B cells was clearly increased in CTRL B cells upon BTK overexpression showcasing that increase signaling results in increase generation of λLC B cells. In the absence of Kidins220, enforced expression of BTK slightly increased the proportion of λLC B cells, but failed to rescue to CTRL levels. BTK overexpression also slightly increased the amount of κLC B cells in both CTRL and B KO cells. This is in line with previously published data showing that BTK is mainly needed for the generation of λLC B cells (*Dingjan et al., 2001; Kersseboom et al., 2006; Xu et al., 2007*). Taken together, these data support a role of Kidins220 guaranteeing optimal (pre-) BCR signals regulating the development of λLC B cells. These data emphasize once again an additional role of Kidins220 beyond regulating (pre-) BCR signaling via BTK. Based in our data, Kidins220 most probably enhances the survival of developing B cells to ensure that in case of unsuccessful rearrangement of the *κLC* locus, rearrangement of the *λLC* locus can take place.

Alternatively, this question could be addressed genetically by crossing our B-KO mice with mice expressing a constitutive active BTK (E14K mutant). It has been previously reported that the introduction of this mutation increased the proportion of λLC B cells in mice (*Dingjan et al., 2001*). Unfortunately, we did not have access to the BTK-E14K mutant mice. We are confident that the Reviewer´s question is answered by our experiments and that he/she agrees that creating a new mouse model would be beyond the scope of this manuscript.

The new data are displayed in new Figure 9 and lines 509-520 in the transferred manuscript.

3. The authors speculate about a link via reduced activity of RAG and/or E2A proteins, but no experimental data on this is shown. Exploration of the mechanistic aspect would be interesting and if such a mechanism would be identified that would clearly lift the impact of the work. However, the task is not easy and might not be fast to solve.

We thank the Reviewer for this comment and for acknowledging that the task is indeed challenging. We have addressed this task twofold: by directly addressing RAG activity and by analyzing *Rag1/2* expression levels.

Firstly, we have analyzed RAG1/2 activity by retrovirally transducing IL-7 BM cultures of

CTRL and B-KO mice with a previously described RAG-activity reporter plasmid (GFPi) (*Trancoso et al., 2013*). The plasmid is designed to express two fluorescent proteins: RFP and GFP. On the one hand, the sequence of RFP is in-frame in the plasmid and serves as fluorescent marker for transduction efficiency. On the other hand, the sequence of GFP is inverted and flaked by RAG-recognition signal sequences (RSS). RAG protein activity will lead to a RAG-mediated GFP inversion and thus, a GFP fluorescence signal is expected as readout for RAG activity. The results are displayed in new Figure 2—figure supplement 1D. RAG activity was equally induced in both CTRL and B-KO B cells in BM cultures three days upon IL-7 withdrawal as shown by the comparable percentages of GFP^+^ cells. Based on these results, we conclude that RAG activity *per se* is not affected by the absence of Kidins220 expression. Text editions are in the main text lines 222-231.

Secondly, we have analyzed *Rag1/2* expression levels in two complementary settings: in BM cultures shortly upon IL-7 withdrawal (equivalent conditions to the RAG activity assay), in *ex-vivo* sorted B cells. Both shortly upon IL-7 withdrawal and in sorted developing B cells, the levels of *Rag1* and *Rag2* were comparable between CRTL and B-KO cells (Figure 2—figure supplement 1B and C).

Text editions in the text lines 221-222.

Taken together, *Rag1/2* levels shortly upon IL-7 withdrawal are sufficient for efficient recombination in the absence of Kidins220. This aligns with the normal generation of κLC B cells in B-KO, since it is generally accepted that rearrangement at the *κLC* locus occurs first (*Engel et al., 2001; Luning Prak et al., 2011; Nemazee, 2006*). However, in long-term BM cultures overexpressing BCL2, expression of *Rag2* and *E47* (see below, question #4) significantly diminishes in B-KO cells leading to the subsequent reduction in rearrangement at the *λLC* locus (Figure 4E). It is worth noting that rearrangement at the *λLC* locus demands higher levels of E2A proteins (*Beck et al., 2009*). This further reinforces the notion that in the absence of Kidins220, the levels of RAG and E2A proteins are insufficient for rearranging the *λLC* locus. Unfortunately, we were unable to investigate the activity of E2A proteins using similar approaches. In line with our results of *Rag1/2* expression, our qRT-PCR results showed reduced *E47* expression upon IL-7 withdrawal in long-term BM cultures, while *E12* expression was unaffected (Figure 4E).

Text editions in the main text lines 321-326.

4. Line 278-88. The authors state that the dampened abundance of λ-LC germline transcripts was not caused by an altered expression of, E47, Rag1 and Rag2. However, this seems to be based on only one experiment with one mouse per genotype (Figure S4). The direct comparison of the data is challenged by the genotypes not shown directly next to each other, but there could be a reduction in Rag 2 and E47, which is worth verifying with proper replicates.

Following the Reviewer’s suggestion, we have changed the display of the former Figure S4, which is now Figure 4E and have further strengthened our results by analyzing the expression of *Rag1*, *Rag2* and *Tcf3* encoding for E2A (*E12* and *E47*) in additional biological replicates. Our new results show a statistically significant reduction in the expression levels of *E47* and *Rag2* over time upon IL-7 withdrawal in BM cultures overexpressing BCL2.

We want to use the opportunity to further discuss our data. We did not detected differences in *Rag1/2* or E2A (*E12* and *E47*) expression in directly sorted cells or short-term BM cultures upon IL-7 withdrawal when CTRL and B-KO were compared (Figure 2—figure supplement 1B and C). However, we detected differences at later time points, when *Bcl2* was overexpressed to allow long-term *in-vitro* cultures (Figure 4E). These results might appear conflicting. Taking in consideration our compelling evidence that in the absence of Kidins220 the survival and fitness of developing B cells is reduced, we think that we have already selected for the most robust cells in our primary sorted cells as well as in our short term BM cultures t. Under these conditions, no differences in *Rag1/2* or E2A (*E12* and *E47*) expression were observed because those cells with reduced expression of the transcription factors did not survive. It is only over time when survival is enforced by overexpression of BCL2 that the absence of Kidins220 ultimately impacts recombination at the *λLC* locus by an insufficient induction of *Rag1/2* or *E12/E47* levels (Figure 4E).

We have now rewritten the paragraph in the new version of the manuscript accordingly (321-326).

5. The breeding strategies of the various mouse crosses as well as the principles in the selection of the control animals should explained in the method section.

We have included this information now in our methods part (lines 683-691).

6. line 130-131: "Pattern was confirmed.." However, the % of λ V genes do not seem very similar to the 62%, 31%, 7% in Figure 1B. Maybe this could be revised.

We thank the Reviewer for pointing out that this sentence and the former Figure 1B were misleading.

The ratio of *λLC* V-gene usage in wild type mice is about 62%, 31%, 7% for λ1, λ2 and λ3, respectively with respect to all λLC B cells (*Boudinot et al., 1994; Sanchez et al., 1996*). We have now calculated the percent of the different families in the same way and we obtained 59.5%, 31.8%, and 8.7% for λ1, λ2 and λ3, respectively (new Figure 1D). Thus, our results are in line with the literature and are now represented in new Figure 1D. We have additionally separated the graphs for κLC and λLC in the new version of our manuscript to improve understandability (new Figures 1C and 1D).

7. line 365: The Figure reference should probably state Figure 6K.

We thank the Reviewer for noticing the mistake in line 365. We have now corrected it. Please note that due to a reorganization of the figures, former Figure 6K is now Figure 6— figure supplement 1H.

8. lines 420-1. It is stated that B-KO mice fail to efficiently upregulate MHCII. If this statement is based on the data in the previous paper, that should be explain more clearly. The data shown in Figure 8A shows no difference in MHCII+ population.

We thank the Reviewer for pointing out this issue. Several reports have used the level of expression of MHCII as a read out for efficient pre-BCR signaling during B cell development. Indeed, mutations in well-described signaling molecules downstream of the pre-BCR resulted in reduced levels of MHCII on the surface of developing B cells (*Kersseboom et al., 2003; Kersseboom et al., 2006; Middendorp et al., 2002; Stadhouders et al., 2014; Xu et al., 2007*). For instance, in Figure 2 from *Kersseboom et al., 2003* reduced levels of MHCII were detected in BTK- or SLP65-deficient pre-B cells. Based on these reports, we gated for CD19^+^LC^-^ pre-B cells and observed a clear decrease in MHCII^+^ positive cells in B-KO mice when compared to CTRL littermates. We have now rewritten the text in the new version to avoid future confusions. Furthermore, we have improved the Figure 8A by including more data points.

Please see the editions in the main text in lines 481.

9. Especially considering the wealth of strains and models used in this work, it is important that extra care is taken on the clarity of the figures, as the authors have well done. However, in Figure 3 there is discrepancy of the text using κ-KO abbreviation and the figures stating iEκ+/T and iEκT/T.

We thank the Reviewer for pointing out that the abbreviations of the different mouse models were not consistent in this case. We revised our manuscript and implemented more consistent abbreviations in the new version of the manuscript.

Please find our editions in the main text lines 240-242 and in Figures and figure legends of Figure 3 and Figure 3—figure supplement 1.

Reviewer #2 (Significance (Required)):The experimental set up is very thorough and carefully conducted. The data is robust and the manuscript is carefully prepared, and it provides as an excellent and thorough example on how to systematically inspect the different factors affecting λ chain usage. The idea of Kidins220 regulating λ locus accessibility by modulation of pre-BCR signalling is highly interesting. Unfortunately, that hypothesis remains speculative and could be difficult to solve in a meaningful time frame. However, the phenotype in the λ LC usage is very strong and also interesting as such. The study has novelty and impact to catch audience in the fields of B cell development and tolerance, particularly.My expertise lies in BCR signalling, rather than pre-BCR.

We thank the Reviewer for his/her thoughtful synthesis and for recognizing the relevance of this study. We are confident that the Reviewer will appreciate our efforts in providing new data supporting to the hypothesis that Kidins220 regulates *λLC* locus accessibility by modulating pre-BCR signaling.

Reviewer #3 (Evidence, reproducibility and clarity (Required)):This manuscript investigates the role of Kidins220 in B cell development, focusing on the regulation of λ light chain expression. The authors convincingly demonstrate that in the absence of Kidins220, λ, but not κ, light chain is poorly transcribed. This is not improved by forcing usage of the λ locus by deleting κ or by expressing an autoantigen that deletes κ expressing cells. In part, λ light chain expression in Kidins220-deficient B cells is improved by preventing apoptosis by overexpression of BCl^-^2. The authors also show reduced ROS production during B cell development of Kidins220-deficient B cells. The authors conclude that Kidins220 is required for preBCR or BCR-induced initiation of λ light chain locus directly, as well as prolonging survival of the B cells to afford time for using this chain after κ failed to generate a functional, non-autoreactive BCR.The work is of very high quality and the experiments are well constructed and the data are clear and convincing. There is a great level of detail in the analysis of the BCR repertoire and the effects of Kidins220 on the light chain expression during the various genetic manipulations and the figures are informative and contain all the essential analyses.

We thank the reviewer for his/her precise synthesis, for highlighting our major findings and for his/her positive comments towards our work.

1. While the data are clear, sometimes the description is difficult to follow due to the level of detail. In most of the sections, a clearer focus on the individual hypotheses about how Kidins220 regulates λ chain expression is needed.

We thank the Reviewer for her/his suggestions to improve clarity and get a more straightforward message. We have implemented her/his suggestion along the manuscript.

2. The most open question that is not addressed is what is the signalling pathway by which Kidins220 regulates λ light chain selectively? The previous work of the authors analysed calcium, MAPK and PI3K signalling in the knock out B cells. Here, the manuscript focuses on ROS production. It is not clear whether ROS production is a substitute for "general" strength of BCR signalling and therefore the effects on the λ locus are a matter of quantitatively reduced signalling, or whether some specific pathways are more affected by Kidins220 deficiency than others and may help to explain the λ phenotype. Is it possible to look at MAPK, PI3K and NFkB signalling? How about signalling to transcription factors that were implicated in λ locus function such as IRF4? Providing just a little more information would link what is known about the signalling functions of Kidins220 with the in vivo phenotypes and would enhance the overall value of the work.

We thank the Reviewer for his/her suggestion to provide deeper insights into signaling network by which Kidins220 regulates λLC expression. This point is in line with the requests expressed by the other two Reviewers. Therefore, we will like to refer to the response to question #6 formulated by Reviewer #1 for a detailed answer. Briefly, we have investigated (pre-) BCR signaling in the presence and absence of Kidins220 by analyzing the levels of phospho-proteins using phospho-flow. Our results are summarized here. Firstly, we did not detect differences in proximal (pre-) BCR signaling (phospho-SYK and phospho-SLP65, new Figure 8B). Secondly, we did not detect major differences in respect to NF-κB signaling between B-KO and CTRL B cells (new Figure 8C). Lastly, reduced levels of phospho-ERK were observed in both pre-B and immature B cells lacking Kidins220 (new Figure 8D).

In addition, we observed enhanced phospho-AKT and phospho-FOXO1 levels in the absence of Kidins220 in pro- and pre-B cells indicative of increased signaling via the IL-7R since IL-7R levels were comparable in the presence and absence of Kidins220 (new Figure 4—figure supplement 1).

Taken together, our data point towards a role of Kidins220 in supporting (pre-) BCR signaling via ROS and ERK, as well as IL-7R signaling via AKT/FOXO1. The absence of Kidins220 notably affects developmental stages in which the B cells are highly dependent on signals supporting survival and V(D)J recombination. Hence, λLC B cells are the most affected by the absence of Kidins220.

Our editions are in the main text lines 300-307 and 482-489, as well as in the new Figure 8B-D and Figure 4—figure supplement 1B and C.

3. In some figures, the numbers for the λ gene are too small compared to κ to be appreciated. Putting them on a separate graph or using a separate y axis would help. For example Figure 1B.

We thank the Reviewer suggesting these changes to increase the readability of our figures. We have now displayed the two LCs in separate graphs (new Figures 1C-D).

Reviewer #3 (Significance (Required)):The manuscript's main contribution is about the regulation of λ light chain expression during B cell development. In itself, this does not have a broad significance, particularly because the manuscript does not make it clear why Kidins220 is a key molecule in this process in comparison to other BCR-dependent (or BCR-independent) signals. However, the regulation of light chain expression and editing is an important step in B cell development and the decisions that regulate it are important for the generation of normal BCR repertoire. Since this process is defective in various immune deficiencies and in immune dysregulation including autoimmunity, understanding the role of Kidins220 can potentially shine some light on these conditions.

We thank the Reviewer for his/her thoughtful synthesis and for recognizing the relevance of this study. We are confident that the Reviewer will appreciate our efforts in providing new data to clarify why Kidins220 is a key molecule regulating λLC expression and the generation of λLC*-*bearing B cells.

References

Beck K, Peak MM, Ota T, Nemazee D, Murre C. 2009. Distinct roles for E12 and E47 in B cell specification and the sequential rearrangement of immunoglobulin light chain loci. *The Journal of experimental medicine* 206:2271–2284. doi: 10.1084/jem.20090756.

Boudinot P, Drapier AM, Cazenave PA, Sanchez P. 1994. Conserved distribution of λ subtypes from rearranged gene segments to immunoglobulin synthesis in the mouse B cell repertoire. *European Journal of Immunology* 24:2013–2017. doi: 10.1002/eji.1830240912.

Clark MR, Mandal M, Ochiai K, Singh H. 2014. Orchestrating B cell lymphopoiesis through interplay of IL-7 receptor and pre-B cell receptor signalling. *Nature Reviews Immunology* 14:69–80. doi: 10.1038/nri3570.

Dingjan GM, Middendorp S, Dahlenborg K, Maas A, Grosveld F, Hendriks RW. 2001. Bruton's tyrosine kinase regulates the activation of gene rearrangements at the λ light chain locus in precursor B cells in the mouse. *The Journal of experimental medicine* 193:1169–1178. doi: 10.1084/jem.193.10.1169.

Engel H, Rühl H, Benham CJ, Bode J, Weiss S. 2001. Germ-line transcripts of the immunoglobulin λ J–C clusters in the mouse: characterization of the initiation sites and regulatory elements. *Molecular Immunology* 38:289–302. doi: 10.1016/S01615890(01)00056-6.

Feng Y-Y, Tang M, Suzuki M, Gunasekara C, Anbe Y, Hiraoka Y, Liu J, Grasberger H, Ohkita M, Matsumura Y, Wang J-Y, Tsubata T. 2019. Essential Role of NADPH OxidaseDependent Production of Reactive Oxygen Species in Maintenance of Sustained B Cell Receptor Signaling and B Cell Proliferation. *Journal of immunology (Baltimore, Md. 1950)* 202:2546–2557. doi: 10.4049/jimmunol.1800443.

Fiala GJ, Janowska I, Prutek F, Hobeika E, Satapathy A, Sprenger A, Plum T, Seidl M, Dengjel J, Reth M, Cesca F, Brummer T, Minguet S, Schamel WWA. 2015.

Kidins220/ARMS binds to the B cell antigen receptor and regulates B cell development and activation. *The Journal of experimental medicine* 212:1693–1708. doi: 10.1084/jem.20141271.

Goodnow CC. 1996. Balancing immunity and tolerance: deleting and tuning lymphocyte repertoires. *Proceedings of the National Academy of Sciences of the United States of America* 93:2264–2271. doi: 10.1073/pnas.93.6.2264.

Kersseboom R, Middendorp S, Dingjan GM, Dahlenborg K, Reth M, Jumaa H, Hendriks RW. 2003. Bruton's tyrosine kinase cooperates with the B cell linker protein SLP-65 as a tumor suppressor in Pre-B cells. *Journal of Experimental Medicine* 198:91–98. doi: 10.1084/jem.20030615.

Kersseboom R, van Ta BT, Zijlstra AJE, Middendorp S, Jumaa H, van Loo PF, Hendriks RW. 2006. Bruton's tyrosine kinase and SLP-65 regulate pre-B cell differentiation and the induction of Ig light chain gene rearrangement. *Journal of immunology (Baltimore, Md. 1950)* 176:4543–4552. doi: 10.4049/jimmunol.176.8.4543.

Luning Prak ET, Monestier M, Eisenberg RA. 2011. B cell receptor editing in tolerance and autoimmunity. *Annals of the New York Academy of Sciences* 1217:96–121. doi: 10.1111/j.1749-6632.2010.05877.x.

Middendorp S, Dingjan GM, Hendriks RW. 2002. Impaired precursor B cell differentiation in Bruton's tyrosine kinase-deficient mice. *Journal of immunology (Baltimore, Md. : 1950)* 168:2695–2703. doi: 10.4049/jimmunol.168.6.2695.

Miho E, Roškar R, Greiff V, Reddy ST. 2019. Large-scale network analysis reveals the sequence space architecture of antibody repertoires. *Nature communications* 10:1321. doi: 10.1038/s41467-019-09278-8.

Nemazee D. 2006. Receptor editing in lymphocyte development and central tolerance. *Nature Reviews Immunology* 6:728–740. doi: 10.1038/nri1939.

Reth M, Hämmerling GJ, Rajewsky K. 1978. Analysis of the repertoire of anti-NP antibodies in C57BL/6 mice by cell fusion. I. Characterization of antibody families in the primary and hyperimmune response. *European Journal of Immunology* 8:393–400. doi: 10.1002/eji.1830080605.

Sanchez P, Rueff-Juy D, Boudinot P, Hachemi-Rachedi S, Cazenave PA. 1996. The λ B cell repertoire of kappa-deficient mice. *International Reviews of Immunology* 13:357– 368. doi: 10.3109/08830189609061758.

Stadhouders R, Bruijn MJW de, Rother MB, Yuvaraj S, Ribeiro de Almeida C, Kolovos P, van Zelm MC, van Ijcken W, Grosveld F, Soler E, Hendriks RW. 2014. Pre-B cell receptor signaling induces immunoglobulin κ locus accessibility by functional redistribution of enhancer-mediated chromatin interactions. *PLoS Biology* 12:e1001791. doi: 10.1371/journal.pbio.1001791.

Trancoso I, Bonnet M, Gardner R, Carneiro J, Barreto VM, Demengeot J, Sarmento LM. 2013. A Novel Quantitative Fluorescent Reporter Assay for RAG Targets and RAG Activity. *Frontiers in Immunology* 4:110. doi: 10.3389/fimmu.2013.00110.

Tsubata T. 2020. Involvement of Reactive Oxygen Species (ROS) in BCR Signaling as a Second Messenger. *Advances in experimental medicine and biology* 1254:37–46. doi: 10.1007/978-981-15-3532-1_3.

Wheeler ML, Defranco AL. 2012. Prolonged production of reactive oxygen species in response to B cell receptor stimulation promotes B cell activation and proliferation. *Journal of immunology (Baltimore, Md. : 1950)* 189:4405–4416. doi: 10.4049/jimmunol.1201433.

Xu S, Lee K-G, Huo J, Kurosaki T, Lam K-P. 2007. Combined deficiencies in Bruton tyrosine kinase and phospholipase Cgamma2 arrest B-cell development at a pre-BCR+ stage. *Blood* 109:3377–3384. doi: 10.1182/blood-2006-07-036418.